# Sigh generation in preBötzinger complex

**Yan Cui[1,2†], Evgeny Bondarenko[2†], Carolina Thörn Perez[2,3], Delia N Chiu[2,4], Jack L Feldman[2*]**

[1]Department of Physiology, Chengdu Medical College, Chengdu, China; [2]Department of Neurobiology, DGSOM, UCLA, Los Angeles, United States; [3]Gene Expression Laboratory, Salk Institute for Biological studies, La Jolla, United States; [4]Synaptic Physiology and Plasticity Group, European Neuroscience Institute Göttingen - A Joint Initiative of the University Medical Center Göttingen and the Max-Planck Society, Göttingen, Germany

## eLife Assessment

This **valuable** study by Cui et al. investigates mechanisms generating sighs, which are crucial for respiratory function and linked to emotional states. Utilizing advanced methods in mice, they provide **solid** evidence that increased excitability in specific preBötzinger complex neuronal subpopulations expressing Neuromedin B receptors, gastrin-releasing peptide receptors, or somatostatin can induce sigh-like large amplitude inspirations. With additional technical clarifications and further elaboration of the limitations in terms of how the results are interpreted in the revised manuscript, the study will interest neuroscientists studying respiratory neurobiology and rhythmic motor systems.

**\*For correspondence:**
feldman@g.ucla.edu

†These authors contributed equally to this work

**Competing interest:** The authors declare that no competing interests exist.

**Abstract** We explored neural mechanisms underlying sighing in mice. Photostimulation of parafacial (pF) neuromedin B (NMB) or gastrin-releasing peptide (GRP), or preBötzinger Complex (preBötC) NMBR or GRPR neurons elicited ectopic sighs with latency inversely related to time from preceding endogenous sigh. Of particular note, ectopic sighs could be produced without involvement of these peptides or their receptors in preBötC. Moreover, chemogenetic or optogenetic activation of preBötC SST neurons induced sighing, even in the presence of NMBR and/or GRPR antagonists. We propose that an increase in the excitability of preBötC NMBR or GRPR neurons not requiring activation of their peptide receptors activates partially overlapping pathways to generate sighs, and that preBötC SST neurons are a downstream element in the sigh generation circuit that converts normal breaths into sighs.

## Introduction

Endogenous sighs generated periodically, typically every few minutes in rodents and humans, reinflate collapsed alveoli to maintain proper gas exchange in the mammalian lung (*Knowlton and Larrabee, 1946*; *Mccutcheon, 1953*; *Reynolds, 1962*). Two largely parallel medullary pathways from the pF to preBötC appear critical for the generation of these physiological sighs: pF neurons expressing NMB or GRP project to preBötC neurons expressing their cognate receptors, NMBR and GRPR (*Li et al., 2016*). These pathways also appear to mediate confinement-induced claustrophobic sighing (*Li et al., 2020*), and by extension, other forms of sighing associated with emotional states such as during sadness, anxiety, depression, relief, or happiness (*Ramirez, 2014*; *Li and Yackle, 2017*).

In rodents, the preBötC receives inputs from NMB and GRP pF neurons, and genetic deletion of NMBRs or GRPRs or local antagonism of NMBRs or GRPRs in preBötC significantly reduces endogenous sighing, while local injection into preBötC of bombesin, NMB or GRP significantly increases spontaneous sighing (*Li et al., 2016*). In order to determine whether NMB and GRP are obligatory for

sighs, and to better understand the mechanisms within the preBötC underlying sigh generation, we investigated whether activation of pF NMB or GRP neurons can generate sufficient input to preBötC to induce sighs and whether activation of preBötC GRPRs or NMBRs is necessary for generation of sighs.

Using transgenic mice that express Cre- or Flp-recombinase in *Nmb-*, *Grp-*, *Nmbr-*, or *Grpr-*expressing neurons, we investigated the effects on sighing and other aspects of breathing pattern in vivo: (i) of optogenetic activation of ChR2-transfected pF GRP or NMB neurons; (ii) of optogenetic or chemogenetic activation of ChR2-transfected preBötC GRPR or NMBR neurons, including in the presence of their antagonists, and; (iii) determined whether activating preBötC SST neurons, presumptive preBötC output neurons, can generate sighs. Moreover, by using electrophysiology and two-photon calcium imaging, we investigated the activity profiles of *Grpr-* or *Nmbr-*expressing preBötC neurons in vitro.

We conclude that: (i) activation of either pF NMB or GRP neurons generates sufficient input to preBötC to generate ectopic sighs; (ii) sighs can be generated by activation of preBötC GRPR and NMBR neurons even after blockade of NMBR and GRPR receptors; (iii) preBötC GRPR or NMBR neurons act via partially overlapping pathways to produce sighs via preBötC SST neurons; (iv) preBötC NMBR neurons are not rhythmogenic, and; (v) activation of preBötC SST neurons can generate sighs.

## Results

A sigh is an inspiratory effort that results in significantly increased tidal volume ($V_T$), typically two to five times larger compared to normal breaths with a range of airflow profiles. Here, we define sighs as inspiratory efforts that result in transient significantly (> twofold) increased $V_T$, which occur periodically at intervals longer than eupneic, i.e., normal, breaths. For all analyses, we designated sighs by their larger $V_T$ irrespective of their shape. Thus, monophasic 'augmented breaths,' biphasic 'partial doublets,' or 'full doublets' were all considered sighs (*Appendix 1—figure 1*). The mechanisms underlying formation of these different sigh subtypes are outside the scope of this publication.

### Activation of *Nmb-* or *Grp-*expressing pF neurons induces sighs

To test whether selective activation of pF NMB or GRP neurons, i.e., bombesin peptide-expressing pF neurons generates sighs, we established *Grp*-ChR2 or *Nmb*-ChR2 mouse lines (see *Methods*) by crossing floxed-ChR2-tdTomato mice with *Grp*^Cre and *Nmb*^Cre mice (*Figure 1a and c*). We then measured the effects of targeted pF photostimulation (long pulse photostimulation, LPP, single bilateral 4–10 s 5 mW pulse or short pulse photostimulation, SPP, single bilateral 100–500 ms pulse; see *Methods*) *Cui et al., 2016* in anesthetized *Grp*-ChR2 or *Nmb*-ChR2 mice.

In brainstems of adult *Grp*-ChR2 mice at the rostrocaudal level of the facial nucleus (7 N), tdTomato fluorescence is expressed mainly in the dorsomedial part of retrotrapezoid nucleus/parafacial respiratory group (RTN/pFRG; *Figure 1—figure supplement 1a*), consistent with previous data in newborn mice (*Li et al., 2016*). In brainstems of adult *Nmb*-ChR2 mice, *Nmb*-tdTomato is mainly expressed ventral and lateral to 7 N, with a few of these neurons in the 7 N, consistent with previous data in mice (*Shi et al., 2017*; *Figure 1—figure supplement 1b*).

In *Grp*-ChR2 mice, LPP in pF elicited a large breath equivalent to an endogenous sigh in both amplitude and tidal volume ($V_T$) when initiated during phase 0.3–1.0 of the sigh cycle (see *Methods*); we refer to such events as ectopic sighs (*Figure 1a*, bottom). Ectopic sighs in *Grp*-ChR2 mice were of partial doublet or augmented breath shape (See *Appendix 1—figure 1*) and were indistinguishable from endogenous sighs in both $V_T$ and inspiratory ($T_I$) and expiratory ($T_E$) duration and exhibited a postsigh apnea ($T_E$ immediately following ectopic sighs increased to 216 ± 35% of $T_E$ of eupneic breaths, *Figure 1b*). During sigh phase 0.3–0.9, the interval from the previous endogenous sigh to the LPP-induced ectopic sigh, i.e., the perturbed sigh cycle, was shorter than the prior control sigh cycle. The interval between an ectopic sigh and the next (endogenous) sigh was, on average, indistinguishable from the interval between endogenous sighs, indicating that ectopic sighs reset the sigh cycle (*Figure 1a*). No ectopic sighs could be generated by bilateral pF SPP in *Grp*-ChR2 mice during inspiration or expiration (phase: 0.0–1.0).

In *Nmb*-ChR2 mice, the effects of pF SPP and LPP were similar to those in *Grp*-ChR2 mice, as SPP (phase: 0.0–1.0) had no effect on sighing while LPP induced ectopic sighs, which appeared as a

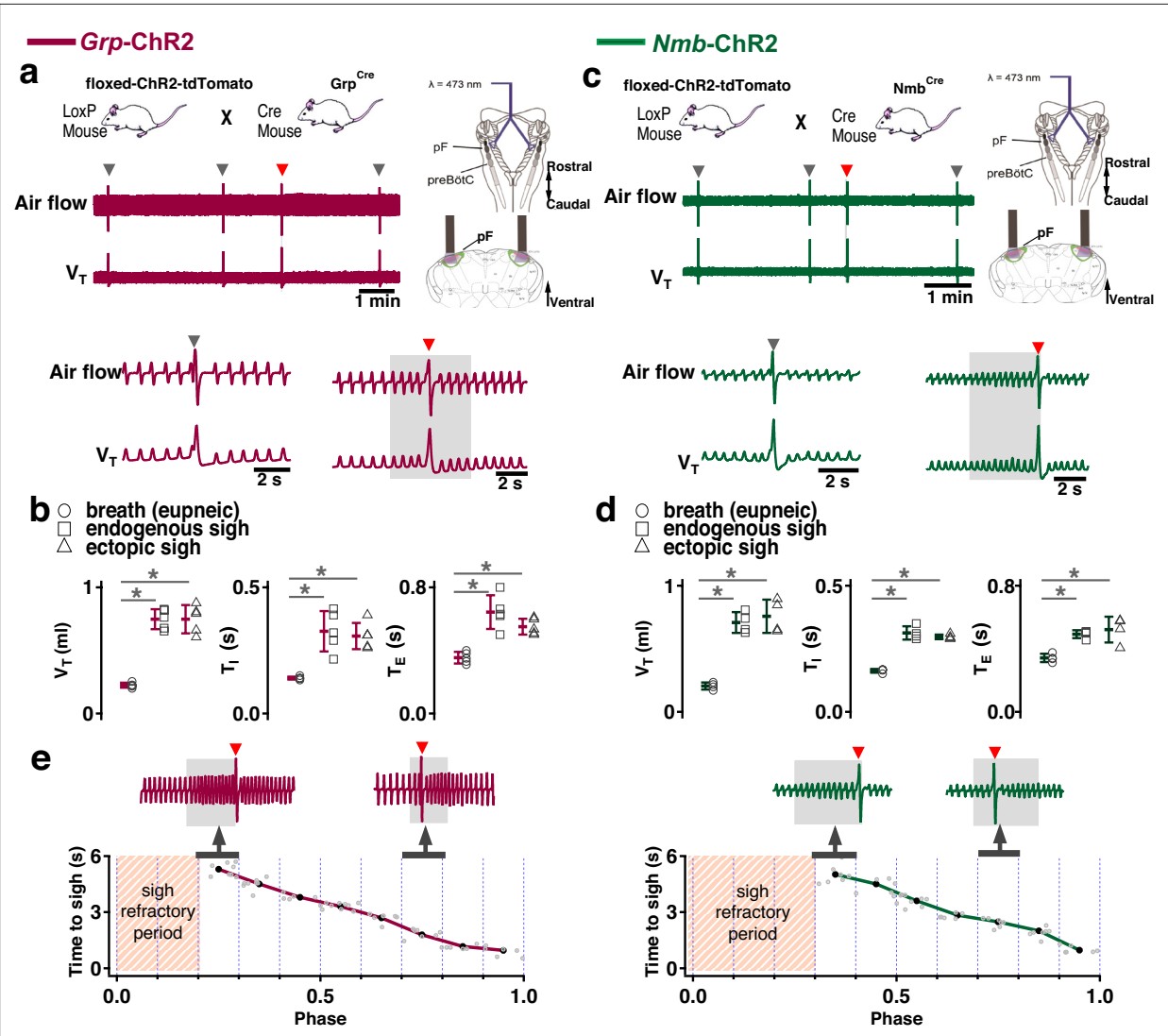

**Figure 1.** Photostimulation of neuromedin B (NMB) and gastrin-releasing peptide (GRP) parafacial (pF) neurons evoked sighing. (**a**) Top: Breeding scheme to generate *Grp*-ChR2 mice. Schematic (right) depicting bilateral placement of optical cannula targeting pF. Middle and Bottom: Raw (middle) and expanded (bottom) traces show bilateral pF long pulse photostimulation (LPP) (right gray box) in *Grp*-ChR2 mice elicits an ectopic sigh (red arrowhead) that appears similar to endogenous sighs (gray arrowheads). (**b**) $V_T$ ($t_3$=0.920, p=0.393), $T_I$ ($t_3$=1.811, p=0.120), and $T_E$ ($t_3$=0.277, p=0.791) of ectopic sighs from pF *Grp*-ChR2 photostimulation were no different from endogenous sighs (n=4 mice), but increased compared to eupneic breaths ($V_T$: $t_4$=10.163, p=5 × 10⁻⁵ $T_I$: $t_3$=6.224, p=7 × 10⁻⁴, $T_E$: $t_4$=7.152, p=4 × 10⁻⁴), indicative of augmented breaths with postsigh apneas. Statistical significance was determined with a One-Way RM ANOVA followed by All Pairwise Multiple Comparison Procedures (Holm-Sidak method), $V_T$: $F_{2,9}$ = 63.194, p<0.001; $T_I$: $F_{2,9}$ = 35.529, p<0.001; $T_E$: $F_{2,9}$ = 32.828, p<0.001. (**c**) Top: Breeding scheme to generate *Nmb*-ChR2 mice. Schematic (right) depicting bilateral placement of optical cannula targeting pF. Middle and Bottom: Raw (middle) and expanded (bottom) traces show bilateral pF LPP (right gray box) in *Nmb*-ChR2 mice elicits ectopic sighs (red arrowhead) that appear similar to endogenous sighs (gray arrowheads). (**d**) $V_T$ ($t_3$=2.436, p=0.0508), $T_I$ ($t_3$=0.603, p=0.569), and $T_E$ ($t_3$=0.308, p=0.768) of ectopic sighs from pF *Nmb*-ChR2 photostimulation were no different from endogenous sighs (n=4 mice), but increased compared to eupneic breaths ($V_T$: $t_3$=37.257, p=3 × 10⁻⁸, $T_I$: $t_3$=6.697, p=5 × 10⁻⁴, $T_E$: $t_3$=7.074, p=4 × 10⁻⁴), indicative of augmented breaths with postsigh apneas. Statistical significance was determined with a One-Way RM ANOVA followed by All Pairwise Multiple Comparison Procedures (Holm-Sidak method), $V_T$: $F_{2,9}$ = 868.829, p<0.001; $T_I$: $F_{2,9}$ = 27.450, p<0.001; $T_E$: $F_{2,9}$ = 34.883, p<0.001. (**e**) Sigh latency from laser onset was negatively correlated with sigh phase in *Grp*-ChR2 (r=–0.995, p=3 × 10⁻⁷) and *Nmb*-ChR2 (r=–0.993, p=8 × 10⁻⁶) mice (Pearson Product Moment Correlation). Top: Representative traces showing that latency between laser onset (gray box indicates LPP) and ectopic sighs (red arrowheads) was longer when LPP was applied just after the refractory period (pink diagonal stripes). Gray dots represent latency to ectopic sigh from stimulation onset in each phase in *Grp*-ChR2 (n=5 mice) and *Nmb*-ChR2 (n=4 mice) mice, solid lines join the average latency in each phase (black circle). No sighs were generated by LPP in sigh phase 0.0–0.2 (34±5 s from previous sigh, measured from 30 stimuli in 5 mice: basal sigh rate 21±3 /hr, range 17–25; intersigh interval 172±25 s) in *Grp*-ChR2 and in phase 0.0–0.3 (58±20 s from previous sigh, measured from 48 stimulus in 4 mice: basal sigh rate 20±7 /hr, range 13–27; intersigh interval 193±65 s) in

*Figure 1 continued on next page*

*Figure 1 continued*

*Nmb*-ChR2 mice. Data are shown as mean ± SE. Asterisks indicate post-hoc multiple comparison test results or paired t-test results: *, significance with p<0.05.

The online version of this article includes the following figure supplement(s) for figure 1:

**Figure supplement 1.** Immunohistochemical verification of ChR2 expression in pF GRP and NMB neurons.

**Figure supplement 2.** Control experiments demonstrating specificity of photostimulation in *Grp*-ChR2 and *Nmb*-ChR2 mice.

partial doublet or augmented breaths, and reset the sigh cycle during 0.4–0.9 sigh phase (*Figure 1c*). Ectopic sighs were also indistinguishable from spontaneous sighs in $V_T$, $T_E$, and $T_I$ and they exhibited a postsigh apnea ($T_E$ after evoked sighs increased to 151 ± 22% of $T_E$ of eupneic breaths, *Figure 1d*).

Notably, in either *Grp*-ChR2 or *Nmb*-ChR2 mice, pF LPP did not elicit ectopic sighs when applied very shortly after an endogenous sigh (phase range: 0.0–0.2 and 0.0–0.3 of endogenous sigh cycle, respectively; *Figure 1e*), indicating a postsigh refractory period for sigh induction. LPP in these cases had no subliminal effect on the sigh cycle, as the expected time to the next endogenous sigh was unchanged.

When pF LPP was initiated immediately following the end of the postsigh refractory period, the perturbed sigh cycle was significantly shorter than the control sigh cycle. For pF LPP later in the sigh cycle, i.e., closer to when the next endogenous sigh was expected, ectopic sighs occurred at even shorter latencies (*Figure 1e*). Thus, after the refractory period during which ectopic sighs could not be elicited, this inverse relationship between photostimulation phase and sigh latency indicated that acute activation of the peptidergic microcircuit was more effective as the sigh phase advanced.

## NMBR and GRPR are expressed primarily in excitatory neurons in preBötC

To understand how the activity of peptide-expressing pF neurons results in the generation of sighs, we next examined expression pattern of *Nmbr*- and *Grpr*-expressing cells in the preBötC using fluorescence in situ hybridization (FISH; *Figure 2*). The vast majority of *Nmbr* and *Grpr* cells (>97%) did not colocalize with the specific astrocytic marker *Aldh1l1*, indicating that these cells are neurons (*Figure 2—figure supplement 1*). *Grpr* neurons were present caudal to pF, in a stream continuing dorsocaudally from pF to preBötC. In contrast, with the exception of sparse *Nmbr/ChAT* neurons in nucleus ambiguus (Amb. ~4 neurons in each section) that were excluded from subsequent analysis, *Nmbr* neurons were mainly found in preBötC. As previously reported (*Li et al., 2016*), some neurons coexpressed *Nmbr* and *Grpr* (*Figure 2b*). In our sample, 57% of *Nmbr* neurons coexpressed *Grpr*, which constituted 32% of the *Grpr* population (*Figure 2b and c*).

Next, we assessed colocalization with the vesicular glutamate transporter *VGlut2* to determine whether *Nmbr* and *Grpr* neurons are glutamatergic. Approximately 85% of *Nmbr* and *Grpr* neurons co-expressed *VGlut2* (*Figure 2b and c*). Finally, we asked whether *Nmbr* and *Grpr* were expressed on preBötC *Sst* neurons. We found scant colocalization of *Sst* and *Nmbr* or *Grpr* (~10%), indicating that the substantial majority of these peptide receptors are not on *Sst* neurons (*Figure 2b and c*).

## Activation of *Nmbr*- or *Grpr*-expressing preBötC neurons induces sighs

Activation of *Nmb*- or *Grp*-expressing pF neurons generates sufficient input to preBötC to generate ectopic sighs, but is activation of neuropeptide-receptor expressing preBötC neurons via their cognate receptors necessary for sigh production? We tested whether bypassing the peptidergic receptors and directly activating preBötC *Grpr*- or *Nmbr*-expressing neurons could also generate sighs. To do this, we expressed ChR2 or the excitatory DREADD hM3Dq in discrete subsets of preBötC neurons (see *Methods*).

To express ChR2 in preBötC GRPR neurons, we microinjected a Flp-dependent ChR2 virus *Fenno et al., 2014* into the preBötC of *Grpr*^Flp^ mice (*Figure 3—figure supplement 1a*). While no ectopic sighs could be generated by bilateral preBötC SPP in these mice (phase: 0.0–1.0), preBötC LPP during 0.2–0.9 sigh phases elicited ectopic sighs which appeared as partial doublet or augmented breaths, and reset the sigh cycle (*Figure 3a*, bottom). Ectopic sighs elicited by preBötC LPP of GRPR neurons were no different from spontaneous sighs in $V_T$ and $T_E$ (*Figure 3b*) and exhibited a postsigh apnea ($T_E$ following evoked sighs increased to 194 ± 22%, *Figure 3b*).

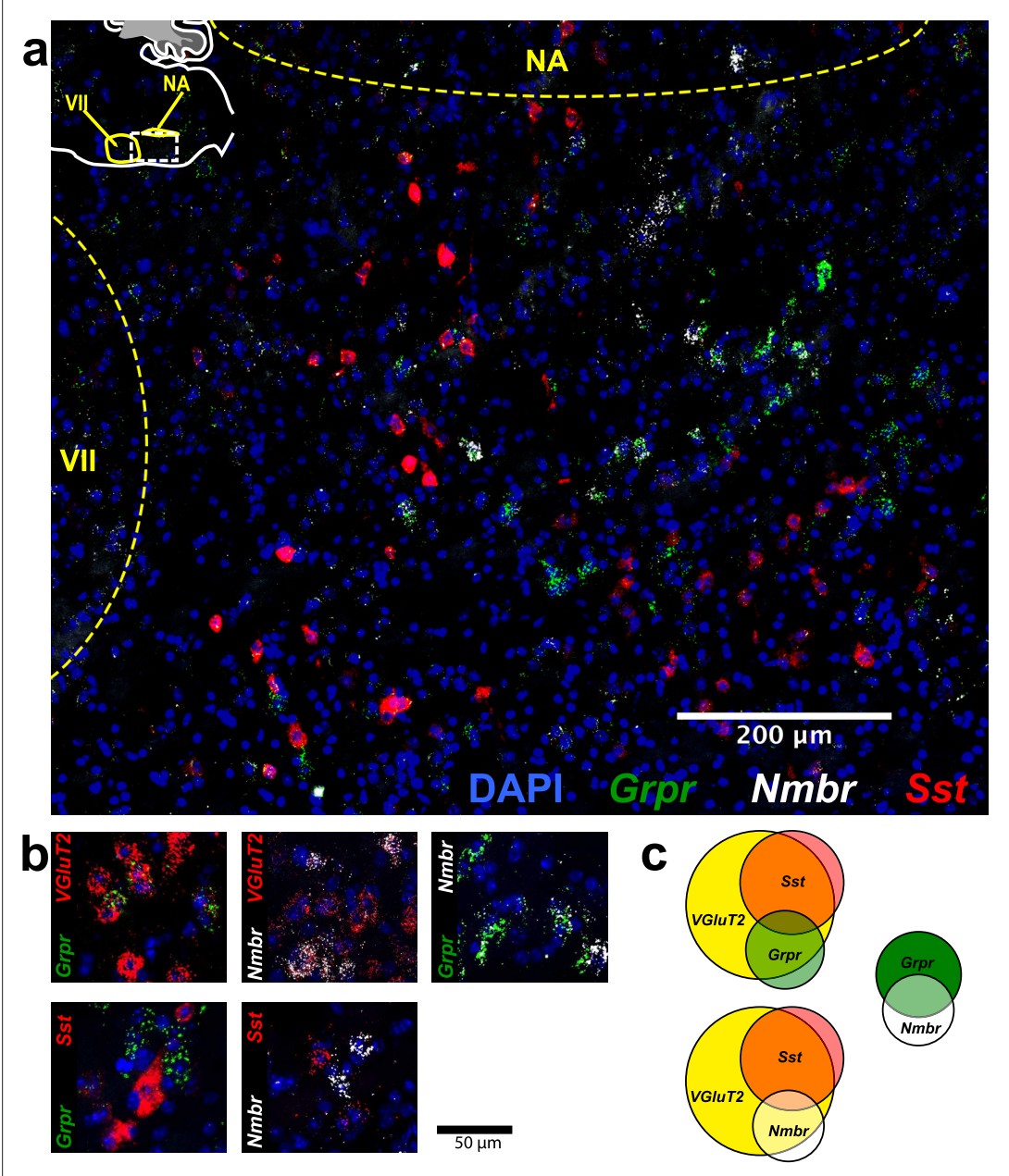

**Figure 2.** *Nmbr* and *Grpr* neurons in preBötzinger Complex (preBötC) are mainly glutamatergic, but mostly not somatostatinergic. (**a**) Sagittal section of medulla with *Nmbr* (white), *Grpr* (green), and *Sst* (red) RNAscope signals as well as DAPI (blue) at the level of preBötC; inset top left: location of image (line box) in mouse medulla. (**b**) Examples of colocalization between relevant markers in high magnification confocal images (100×100 μm) of tissues processed with RNAscope. (**c**), Venn diagrams representing relative number of preBötC neurons expressing relevant markers and their overlap, scaled according to total *Vglut2* count (n=3 sections for each colocalization pair). VII: facial nucleus. NA: nucleus ambiguous. There were 22.6±2.3 *Grpr*+ neurons and 14.6±1.4 *Nmbr*+ neurons in each transverse section; 25/45 *Nmbr*+ neurons coexpressed *Grpr*; 25/77 *Grpr*+ neurons coexpressed *Nmbr*. Majority of both *Nmbr*+ and *Grpr*+ co-expressed *VgluT2* (47/55 and 63/76, respectively), but not *Sst* (6/46 and 5/54, respectively).

The online version of this article includes the following figure supplement(s) for figure 2:

**Figure supplement 1.** RNAscope confirmation of *Nmbr*^Cre and *Grpr*^Flp mouse lines and confirmation of lack of *Nmbr* and *Grpr* expression on astrocites.

Similarly, in *Nmbr*-ChR2 mice (*Figure 3—figure supplement 1b*), preBötC SPP (phase: 0.0–1.0) had no effect on sighing, while LPP during 0.4–0.9 sigh phases elicited ectopic sighs which appeared as a partial doublet or an augmented breath, and reset the sigh cycle (*Figure 3c*, bottom). $V_T$ and postsigh $T_E$ of ectopic sighs were no different from that of spontaneous sighs (*Figure 3d*). Ectopic

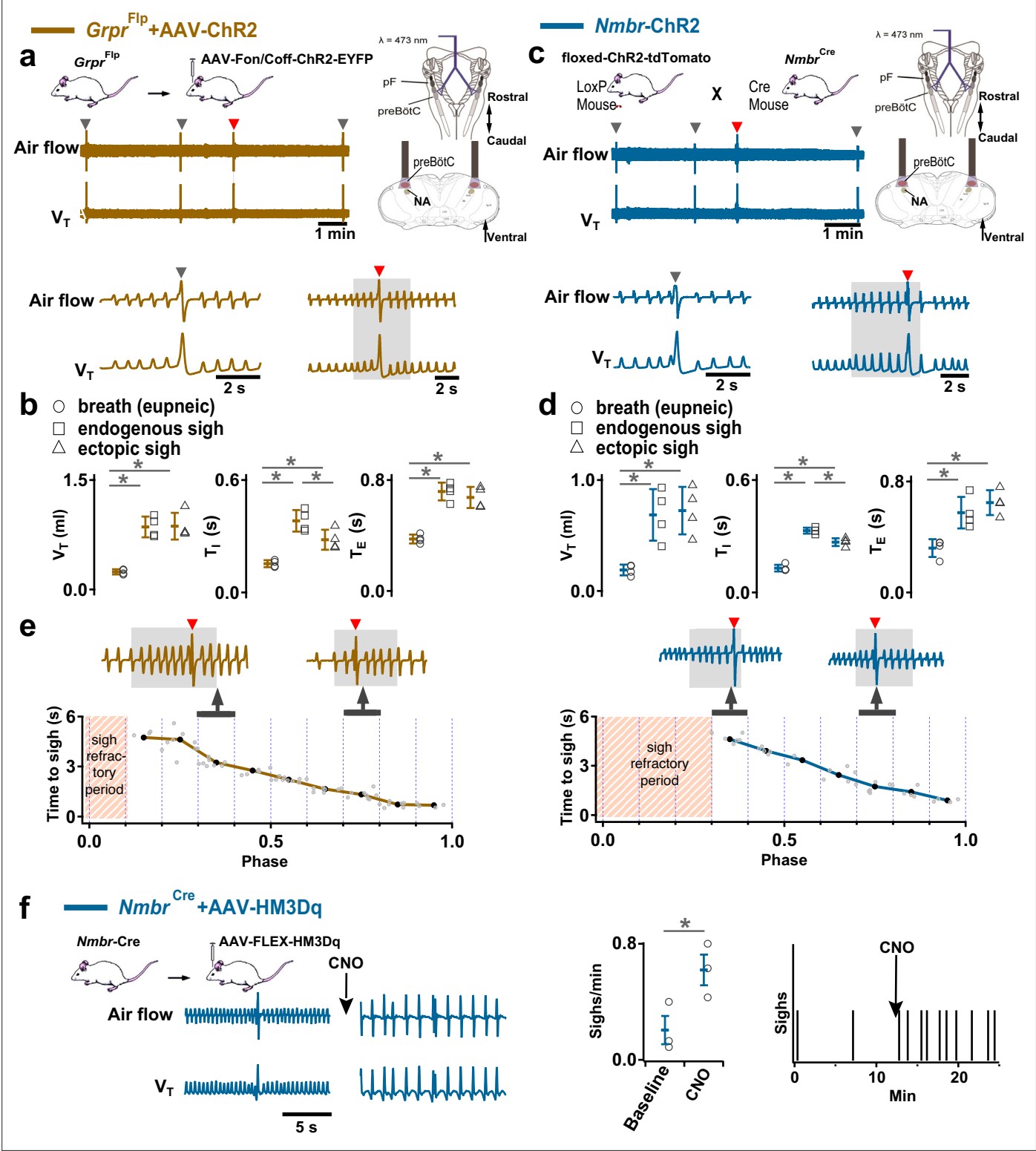

**Figure 3.** Excitation of preBötzinger Complex (preBötC) GRPR or NMBR neurons induced sighs. (**a–e**) Photostimulation of preBötC GRPR (brown) or NMBR (blue) neurons evoked sighs. (**a**) Top: Schematic of genetic strategy to target preBötC GRPR neurons. Schematic (right) depicting bilateral placement of optical cannula targeting preBötC. Middle and Bottom: Raw (middle) and expanded (bottom) traces show bilateral long pulse photostimulation (LPP) (gray box) of preBötC GRPR neurons that could elicit an ectopic sigh (red arrowhead). Gray arrowheads indicate endogenous sighs. (**b**) Photostimulation of preBötC GRPR induced ectopic sighs with similar $V_T$ and $T_E$ as endogenous sighs, but slightly lower $T_I$ ($V_T$: $t_3$=0.233, p=0.824; $T_E$: $t_3$=0.249, p=0.812; $T_I$: $t_3$=3.267, p<0.05). $V_T$, $T_E$, and $T_I$ of ectopic sighs were significantly greater than those of eupneic breaths ($V_T$: $t_3$=6.889,

*Figure 3 continued on next page*

*Figure 3 continued*

p=5 × 10$^{-4}$; $T_I$: t$_3$=3.183, p=0.019; $T_E$: t$_3$=8.675, p=3 × 10$^{-4}$). Statistical significance was determined with a One-Way RM ANOVA followed by All Pairwise Multiple Comparison Procedures (Holm-Sidak method), V$_T$: F$_{2,9}$ = 32.744, p<0.001; $T_I$: F$_{2,9}$ = 16.530, p<0.001; $T_E$: F$_{2,9}$ = 48.771, p<0.001. (c) Top: Targeting scheme to generate *Nmbr*-ChR2 mice. Schematic (right) depicting bilateral placement of optical cannula targeting preBötC. Middle and Bottom: Raw (middle) and expanded (bottom) show bilateral LPP (gray box) of preBötC NMBR neurons could elicit an ectopic sigh (red arrowhead). Gray arrowheads indicate endogenous sighs. (d) Photostimulation of preBötC NMBR neurons induced ectopic sighs with similar V$_T$ and $T_E$ as endogenous sighs, but slightly lower $T_I$ (V$_T$: t$_3$=0.820, p=0.444; $T_E$: t$_3$=0.976, p=0.367; $T_I$: t$_3$=3.175, p=0.0192). V$_T$, $T_E$ and $T_I$ of ectopic sighs were significantly greater than those of eupneic breaths (V$_T$: t$_3$=9.756, p=7 × 10$^{-5}$; $T_I$: t$_3$=7.685, p=3 × 10$^{-4}$; $T_E$: t$_3$=7.326, p=3 × 10$^{-4}$). Statistical significance was determined with a One-Way RM ANOVA followed by All Pairwise Multiple Comparison Procedures (Holm-Sidak method), V$_T$: F$_{2,9}$ = 58.564, p<0.001; $T_I$: F$_{2,9}$ = 62.362, p<0.001; $T_E$: F$_{2,9}$ = 31.646, p<0.001. (e) Latency from laser onset was negatively correlated with phase in *Grpr*$^{Flp}$ (r=–0.980, p=4 × 10$^{-6}$) and *Nmbr*-ChR2 (r=–0.994, p=6 × 10$^{-6}$) mice (Pearson Product Moment Correlation). Representative traces showing latency between laser onset (gray box indicates LPP) and ectopic sighs (red arrowheads) when LPP was applied in medial (0.4) and late (0.8) phase. Gray dots represent latency to ectopic sigh from stimulation onset in each phase in *Grpr*$^{Flp}$ (n=5 mice) and *Nmbr*-ChR2 (n=4 mice) mice, solid lines join the average latency in each phase (black circle). No sighs were generated by LPP in phase 0.0–0.1 (23±4 s from previous sigh, measured from 55 stimulus in 5 mice: basal sigh rate 16±2 hr, range 12–19; intersigh interval 230±41 s) in *Grpr*$^{Flp}$ and in phase 0.0–0.3 (61±21 s from previous sigh, measured from 50 stimulus in 4 mice: basal sigh rate 19±6 hr, range 12–25; intersigh interval 203±70 s) in *Nmbr*-ChR2 mice. (f) Chemogenetic activation of preBötC NMBR neurons induced sighs. Left top: schematic diagram of genetic strategy to selectively express DREADD receptor hM3Dq on preBötC NMBR neurons. Left bottom: representative traces of airflow and V$_T$ before and after application of clozapine-n-oxide (CNO) to brainstem surface. Middle: Activation of hM3Dq receptors expressed on NMBR neurons with CNO significantly increases sigh rate (paired two-tailed t-test, n=3 mice: t$_2$=5.94, p=0.03). Right: Trace from a representative mouse illustrating the incidence of sighs before and after CNO application. Data are shown as mean ± SE. Asterisks indicate post-hoc multiple comparison test results or paired t-test results: *, significance with p<0.05.

The online version of this article includes the following figure supplement(s) for figure 3:

**Figure supplement 1.** Immunohistochemical verification of ChR2 and HM3Dq expression in preBötC GRPR and NMBR neurons.

**Figure supplement 2.** Control applications of clozapine-n-oxide (CNO).

**Figure supplement 3.** Control experiments demonstrating specificity of photostimulation in *Grpr*-Flp and *Nmbr*-ChR2 mice.

sighs elicited by preBötC LPP of NMBR neurons exhibited a postsigh apnea (postsigh $T_E$ increased to 225 ± 33%, *Figure 3d*).

As with photostimulation of pF NMB and GRP neurons, there was a postsigh refractory period during which we could not generate ectopic sighs in preBötC NMBR or GRPR neurons. Again, the latency from LPP to ectopic sigh onset was inversely related to phase (*Figure 3e*).

We also tested whether activation of preBötC NMBR neurons using an excitatory DREADD could induce sighing. To do this, we transduced preBötC *Nmbr*-expressing neurons by microinjection of Cre-dependent AAV-hM3Dq in *Nmbr*$^{Cre}$ mice (*Figure 3—figure supplement 1c*). In these mice, application of the selective agonist clozapine-n-oxide (CNO) to the brainstem surface increased sigh frequency ~threefold (*Figure 3f*; see *Figure 3—figure supplement 2* for control CNO application). Sighs induced by CNO in *Nmbr*$^{Cre}$ mice were of the doublet shape. Thus, increasing the excitability of preBötC NMBR neurons increases sighing.

## preBötC *Nmbr* neurons are rhythmically active

We next asked: what are the electrophysiological properties of the *Grpr*- and *Nmbr*-expressing preBötC neurons? We took several approaches. First, using inspiratory rhythmic slices *Smith et al., 1991* from neonatal *Grpr*-EGFP or *Nmbr*-tdTomato reporter mice, we obtained whole-cell patch-clamp recordings from preBötC GRPR or NMBR neurons while monitoring hypoglossal nerve (XII) inspiratory activity. In *Grpr*-EGFP mice, only 3 of 21 preBötC GRPR neurons were inspiratory-modulated, i.e., fired during XII inspiratory bursts. In contrast, in *Nmbr*-tdTomato mice, the majority of preBötC NMBR neurons (10/16) were inspiratory-modulated (*Figure 4a*). This finding was confirmed using two-photon calcium imaging of rhythmic slices from *Nmbr*-GCaMP6f mice, in which we could monitor the activity of up to 10 neurons simultaneously (*Figure 4b*). The majority of preBötC NMBR neurons (35/58; n=6) exhibited inspiratory-modulated Ca$^{2+}$ oscillations coincident with XII bursts (*Figure 4b*); however, episodes of non-breathing-modulated episodic bursting, i.e., bursts not in phase with XII, were also seen (*Figure 4c*) and varied in frequency from 0.025 to 0.1 Hz (*Figure 4—figure supplement 1*). Almost all NMBR neurons that had Ca$^{2+}$ elevations during inspiratory bursts also had elevations during sighs (34/35), but not during the pre-I period, consistent with whole-cell recordings (*Figure 4a*), indicating that they are likely Type II, i.e., presumptive non-rhythm generating, neurons (*Gray et al.,*

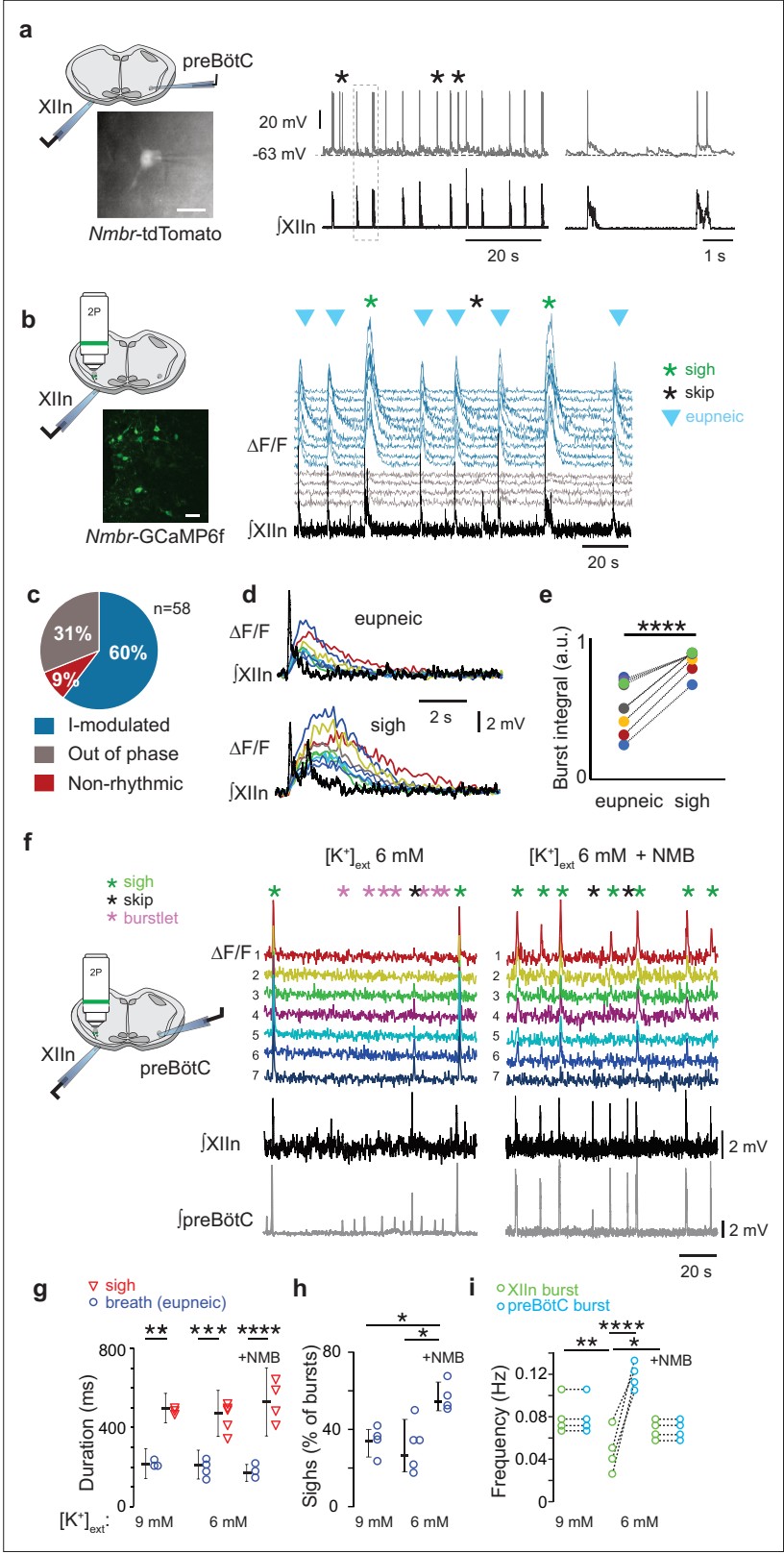

**Figure 4.** NMBR neurons in preBötzinger Complex (preBötC) were rhythmically active in vitro and NMB converted inspiratory burstlets into sighs and bursts. (**a**) Left: schematic of slice preparation with photograph of patched *Nmbr*-tdTomato preBötC neuron. Scale bar: 50 μm. Right: Whole cell current clamp recording from inspiratory-modulated preBötC NMBR neuron (top trace) that generated action potentials (APs) during each XII burst

*Figure 4 continued*

(bottom trace). Not every cluster of APs (*) is associated with a burst, but each burst was associated with one or more Aps. Far right: Two bursts shown at expanded timescale. (**b**) Left: schematic of slice preparation in 2 P experiment with image of GCaMP6f-expressing preBötC neurons. Scale bar 50 μm. Right: Representative $Ca^{2+}$ signal of neighboring preBötC NMBR neurons was correlated with XIIn bursts (9 mM $[K^+]_{ext}$). Large amplitude XIIn bursts represent sighs (green asterisks), XIIn bursts not associated with $Ca^{2+}$ signal indicated with black asterisks. (**c**) Percentage of rhythmically active (I-modulated), non-rhythmic and out-of-phase preBötC NMBR neurons in preBötC. (**d**) $Ca^{2+}$ transients in individual neurons were significantly larger during sighs than during eupneic bursts (paired two-tailed t-test: $t_8$=7.14, p<0.001). (**e**) Intervals of sighs were significantly greater during sighs compared to eupneic bursts (same color-coded neurons as in d). (**f**) Left: schematic of slice preparation and configuration. Right: Simultaneous Ca2 +signal from NMBR neurons (n=7), preBötC field recording, and XIIn activity in control conditions ($[K^+]_{ext}$ = 6 mM) and after addition of NMB (30 nM). Burstlets (red asterisks) are seen in control but not with NMB. (**g**) Quantification of duration of eupneic and sigh bursts in 9 mM $[K^+]_{ext}$ and 6 mM $[K^+]_{ext}$ with or without NMB. Differences in duration between eupneic bursts in different conditions, and between sighs in different conditions were not significant (repeated measures ANOVA, $F_{5,19}$ = 20.1, p<0.001; asterisks indicate significance from Tukey post-hoc tests). (**h**) Percentage of bursts that were sighs during 5 min was significantly higher in 6 mM $[K^+]_{ext}$ +NMB than other conditions (repeated measures ANOVA, $F_{2,9}$ = 7.18, p=0.018). (**i**) Comparisons between XIIn burst frequency and preBötC burst frequency in 9 mM $[K^+]_{ext}$ and 6 mM $[K^+]_{ext}$ with or without NMB. XIIn and preBötC burst frequency were similar in 9 mM $[K^+]_{ext}$ and 6 mM $[K^+]_{ext}$ +NMB, and both conditions differ from 6 mM $[K^+]_{ext}$ alone. There is an increase in preBötC burst frequency in 6 mM [K+]ext due to the burstlets (shown in a), (repeated measures ANOVA, $F_{5,23}$ = 11.57, p<0.001). Data are shown as mean ± SE. Asterisks indicate significance from Tukey post-hoc tests or paired t-tests: *, significance with p<0.05; **, significance with p<0.01; ***, significance with p<0.001.

The online version of this article includes the following figure supplement(s) for figure 4:

**Figure supplement 1.** Examples of $Ca^{2+}$ oscillations in vitro in phase with inspiratory rhythm (blue) and out of phase (gray) in (**a**) 6 mM $[K^+]_{ext}$ (neurons 1–4) and (**b**) 9 mM $[K^+]_{ext}$ (neurons 1–3).

*1999*; *Figure 4d*). The burst duration of the GCaMP6f signal associated with sighs was longer-lasting than those associated with eupneic events in the same neuron (*Figure 4d and e*).

## NMB converts NMBR neurons inspiratory burstlets into sighs and bursts

In typical recording conditions in vitro ($[K^+]_{ACSF}$ = 9 mM), the preBötC population activity burst and the periodic XII discharges are in sync. Under conditions of low excitability ($[K^+]_{ACSF}$ <7 mM), however, in addition to these coincident high-amplitude bursts in preBötC and XII, there are low-amplitude burstlets in preBötC but not in XII (*Smith et al., 1991*). We have hypothesized that the burstlets represent the rhythmogenic kernel (*Smith et al., 1991*; *Feldman and Kam, 2015*; *Ashhad and Feldman, 2020*). NMBR neurons had a Type II firing pattern, i.e., no preinspiratory spiking, suggesting that they are not rhythmogenic.

We reliably observed both burstlets and bursts in population activity when we lowered $[K^+]_{ACSF}$ to 6 mM (*Shi et al., 2017*). When we did so, NMBR neurons were still active during bursts but never during burstlets (*Figure 4f*). When we applied NMB (30 nM), only bursts were observed, while both burst and sigh frequency in both preBötC and XII increased (*Figure 4g*). The duration of sighs and eupneic bursts were similar in $[K^+]_{ACSF}$ = 6 mM vs 9 mM (*Figure 4h*), while the percentage of sighs increased by 30% during NMB (30 nM) application in 6 mM $K^+$ (*Figure 4i*), similar to the increase elicited by 30 nM NMB at 9 mM $K^+$ [4].

## Activating preBötC GRPR- or NMBR-only neurons has distinct effects on sighing

Photoactivation of preBötC GRPR or NMBR neurons can induce sighs. Given that there are three bombesin-related peptide receptor-expressing subpopulations in preBötC, i.e., neurons that express *Nmbr* but not *Grpr* (NMBR-only), neurons that express *Grpr* but not *Nmbr* (GRPR-only) and neurons that express both *Nmbr* and *Grpr*, does activation of neurons expressing one but not the other of these receptors affect sighs differently? We used an intersectional genetic approach to target neurons that express only one of these receptors, excluding double-positive (GRPR/NMBR) neurons (*Fenno et al., 2020*). Thus, using Cre-on/Flp-off and Cre-off/Flp-on viral vectors for the delivery of ChR2 in

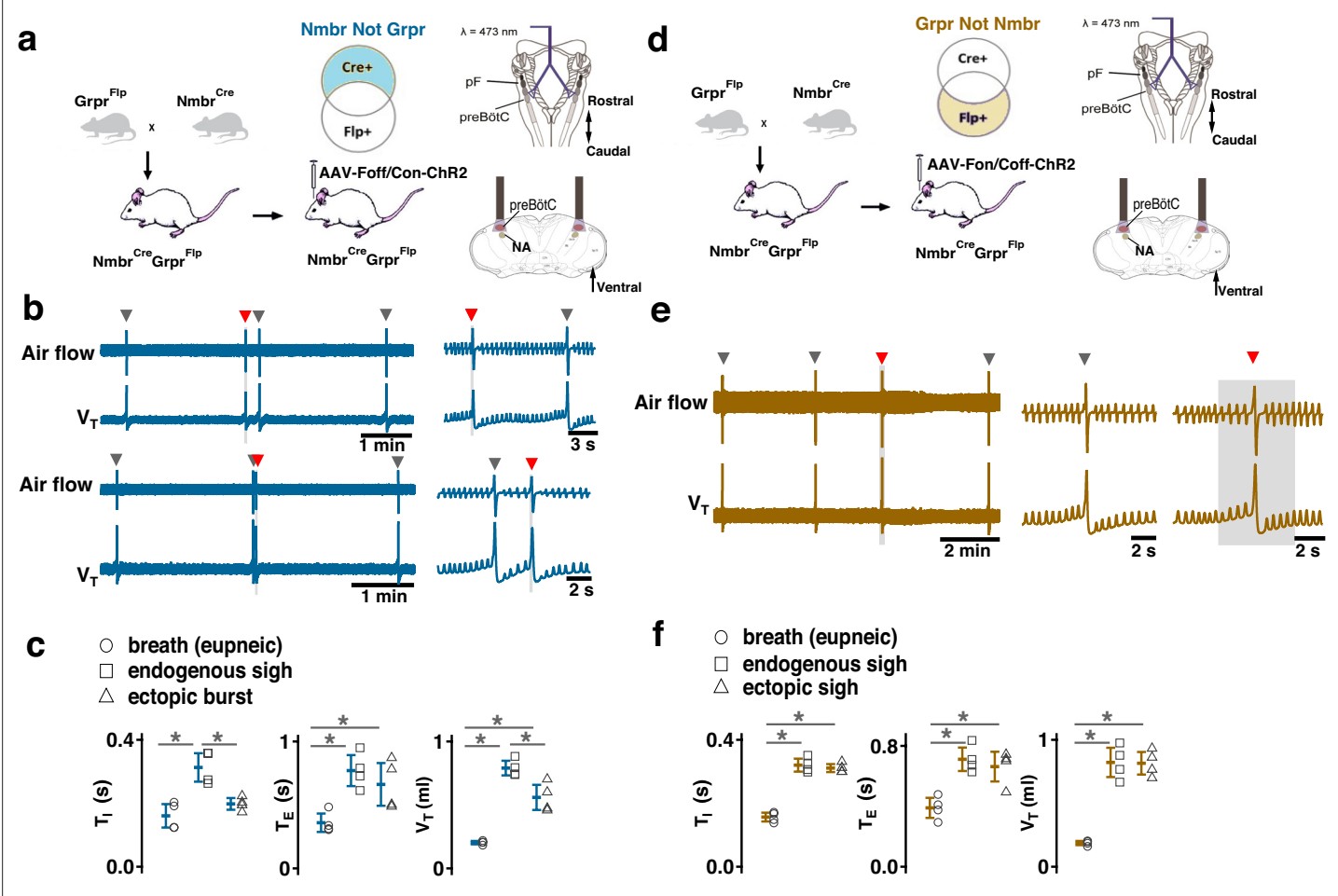

**Figure 5.** Effects of activating preBötzinger Complex (preBötC) NMBR-only or GRPR-only neurons on the generation of sighs. (**a–c**) Effects of activating preBötC NMBR-only neurons on sigh generation. (**a**) Left: Schematic of the intersectional genetic strategy to target preBötC NMBR-only neurons. Right: Schematic depicting bilateral placement of optical cannula targeting preBötC. (**b**) Top: Raw (left) and expanded (right) traces show that preBötC SPP (gray box) of NMBR-only neurons induced an ectopic burst (red arrowhead) which did not reset the sigh rhythm. Bottom: preBötC SPP of NMBR-only neurons induced an ectopic burst (red arrowhead) in a sigh refractory period. Gray arrowheads indicate endogenous sighs. (**c**) $V_T$ of ectopic bursts evoked by NMBR-only photostimulation was smaller than that of endogenous sighs (n=4 mice, $t_3$=4.278, p=0.005), but larger than of eupneic breaths ($t_3$=6.588, p=6 × 10$^{-4}$); $T_E$ of evoked ectopic bursts was no different from endogenous sighs ($t_3$=1.882, p=0.109), but higher than of eupneic breaths ($t_3$=5.203, p=0.002); $T_I$ of evoked ectopic bursts was shorter than that of endogenous sighs ($t_3$=3.917, p=0.008). Statistical significance was determined with a One-Way RM ANOVA followed by All Pairwise Multiple Comparison Procedures (Holm-Sidak method), $V_T$: $F_{2,9}$ = 59.922, p<0.001; $T_I$: $F_{2,9}$ = 25.206, p<0.001; $T_E$: $F_{2,9}$ = 26.939, p<0.001. (**d-f**) Effects of activating preBötC GRPR-only neurons on sigh generation. (**d**) Left: Schematic of the intersectional genetic strategy to target preBötC GRPR-only neurons. Right: Schematic depicting bilateral placement of optical cannula targeting preBötC. (**e**) preBötC LPP of GRPR-only neurons in late phase elicited ectopic sighs and reset the sigh rhythm. Raw (left) and expanded (middle and right) show bilateral long pulse photostimulation (LPP) (gray box) of preBötC GRPR-only neurons elicited an ectopic sigh (red arrowhead). Gray arrowheads indicate endogenous sighs. (**f**) $V_T$ ($t_3$=0.152, p=0.884), $T_I$($t_3$=1.843, p=0.115), and $T_E$ ($t_3$=1.619, p=0.157), of ectopic sighs were no different from endogenous sighs (n=4 mice), but increased compared to eupneic breaths ($V_T$:, $t_3$=12.855, p=10$^{-5}$; $T_I$: $t_3$=30.235, p=9 ×10$^{-8}$; $T_E$: $t_3$=9.007, p=4 × 10$^{-5}$), indicative of augmented breaths with postsigh apneas. Statistical significance was determined with a One-Way RM ANOVA followed by All Pairwise Multiple Comparison Procedures (Holm-Sidak method), $V_T$: $F_{2,9}$ = 111.488, p<0.001; $T_I$: $F_{2,9}$ = 648.855, p<0.001; $T_E$: $F_{2,9}$ = 65.544, p<0.001. Data are shown as mean ± SE. Asterisks indicate post-hoc multiple comparison test or paired t-test results: *, significance with p<0.05.

The online version of this article includes the following figure supplement(s) for figure 5:

**Figure supplement 1.** Immunohistochemical verification of ChR2 expression in preBötC GRPR-only and NMBR-only neurons.

*Nmbr*<sup>Cre</sup>;*Grpr*<sup>Flp</sup> double mutant mice (**Figure 5—figure supplement 1**), we expressed ChR2 in preBötC NMBR-only (**Figure 5a**) or GRPR-only neurons (**Figure 5d**), respectively. We examined the effect of preBötC LPP and SPP of these neurons on sighing. When targeting NMBR-only neurons, preBötC SPP elicited ectopic bursts which appeared similar to partial doublets or augmented breaths with smaller

$V_T$ compared to spontaneous sighs due to a decrease in $T_i$ to 64 ± 8% (**Figure 5c**). $T_E$ immediately following these ectopic bursts were longer than the $T_E$ of eupneic breaths (**Figure 5c**). Note that the time between the SPP-induced ectopic burst and the next endogenous sigh was shorter than the time between endogenous sighs, indicating that SPP-induced burst did not reset the sigh cycle (**Figure 5**, top); SPP-induced ectopic burst could also occur in a sigh refractory period right after the previous eupneic sigh (**Figure 5b**, bottom). When targeting GRPR-only neurons, while ectopic sighs could not be generated by SPP at any phase of the sigh cycle, preBötC LPP in the late phase of the sigh cycle elicited ectopic sighs with a partial doublet or augmented breath shape and reset the sigh cycle (**Figure 5e**). $T_i$, $T_E$, and $V_T$ of LPP-induced sighs were indistinguishable from spontaneous sighs (**Figure 5f**). The ectopic sighs exhibited a postsigh apnea, $T_E$ after evoked sighs increased to 171 ± 12% of the $T_E$ of eupneic breaths (**Figure 5f**).

These data suggest that GRPR and NMBR neurons play different overlapping roles in sigh generation.

## Activating preBötC GRPR or NMBR neurons has distinct effects on inspiratory burst amplitude

In addition to generating sighs, activation of preBötC NMBR and/or GRPR neurons also affected eupneic breathing frequency (*f*) and $V_T$. In *Grpr*$^{Flp}$ mice expressing ChR2 in preBötC, LPP during the sigh refractory period did not generate ectopic sighs but increased *f* to 131 ± 9% of baseline (**Figure 6a**), and increased $V_T$ to 117 ± 4% of baseline (**Figure 6a**), however, when excluding double-positive (GRPR/NMBR) neurons, preBötC SPP or LPP of GRPR-only neurons did not change the inspiratory burst amplitude in stimulus cycles (**Figure 6b**).

In contrast, in *Nmbr*-ChR2 mice, preBötC LPP delivered during a postsigh refractory period did not generate ectopic sighs but decreased *f* to 83 ± 4% (**Figure 6a**) and increased $V_T$ to 140 ± 12% (**Figure 6a**) of baseline. preBötC SPP or LPP of NMBR-only neurons increased $V_T$ to 274 ± 41% or 264 ± 17% of control, respectively (**Figure 6b**).

The effect on inspiratory burst amplitude was also observed in *Nmbr*$^{Cre}$ mice that had preBötC NMBR neurons transfected with Cre-dependent AAV-HM3Dq; CNO applied to the brainstem surface significantly increased $V_T$ by 343 ± 56% (**Figure 6c**) and decreased *f* by 63 ± 8.4% (**Figure 6c**). Note that although such breaths were of the same $V_T$ as eupneic sighs before administration of CNO (**Figure 3c**), in line with our definition of sighs (**Appendix 1—figure 1**), we classify this outcome as large amplitude breaths rather than 'all sigh' breathing rhythm, according to our definition of sighs (**Appendix 1—figure 1**).

Thus, stimulation of preBötC NMBR and/or GRPR neurons also has a profound effect on the eupneic, nonsigh, breathing pattern, and GRPR and NMBR neurons appear to differently affect inspiratory burst amplitude.

## Sighs can be induced after inhibition of GRPRs or NMBRs

Expression of *Grpr* and/or *Nmbr* defines subsets of preBötC neurons that affect sigh generation. Is activation of these peptide receptors *necessary* for sigh generation?

For that, we investigated the effects of activating peptide-expressing pF neurons in the presence of bombesin receptor antagonists. First, we confirmed that bilateral microinjection of the NMBR antagonist BIM23042 and GRPR antagonist RC3095 (300 μM each, 50 nl/side) into the preBötC in anesthetized mice eliminates spontaneous sighs (**Figure 6—figure supplement 1**). Next, we determined the effects of such NMBR and GRPR antagonism on LPP-evoked sighs. After blockade, no sighs were elicited by pF LPP in *Nmb*-ChR2 mice; in contrast, pF LPP of GRP neurons still elicited ectopic sighs of a doublet shape. The $V_T$ was not significantly different from that of spontaneous sighs; $T_E$ of the following respiratory cycle was unaffected (**Figure 6d**). Thus, when pF NMB neurons were stimulated, preBötC NMBR neurons require NMBR activation to contribute to sigh generation, whereas when pF GRP neurons were stimulated, sighs could still be generated after antagonism of preBötC GRPR receptors.

We next investigated whether explicit activation of preBötC NMBRs and/or GRPRs is necessary to produce sighs, i.e., could sighs be generated simply by depolarizing these preBötC neurons to fire action potentials? In ChR2-transduced *Grpr*$^{Flp}$ or *Nmbr*-ChR2 mice after antagonism of both receptors sufficient to eliminate spontaneous sighs (**Figure 6—figure supplement 1**), LPP of preBötC

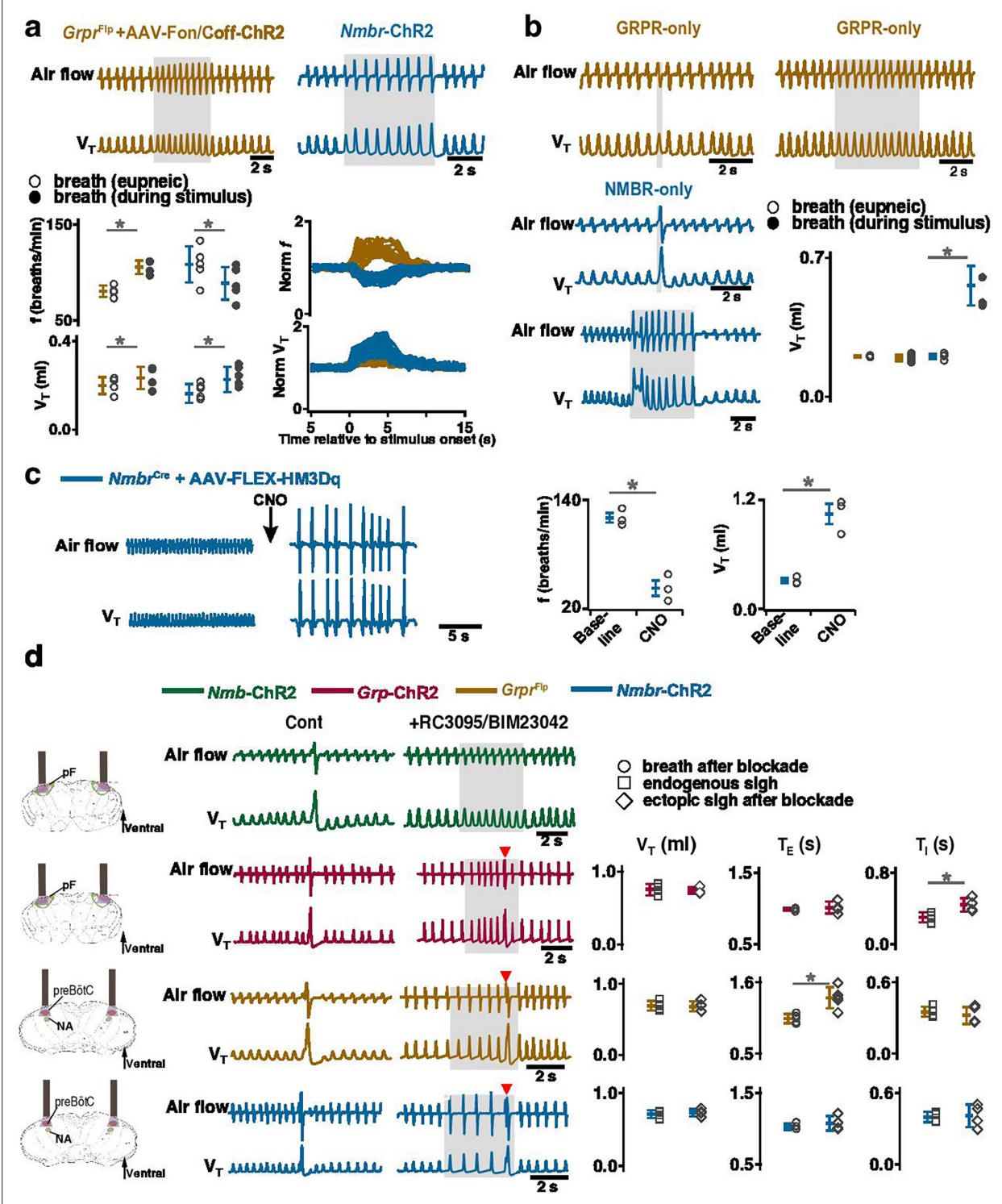

**Figure 6.** Distinct roles of preBötzinger Complex (preBötC) GRPR and NMBR neurons on inspiratory burst amplitude and sigh generation after receptor blockade. (**a**) Photostimulation of preBötC GRPR (brown) and NMBR (blue) neurons perturbs breathing frequency and inspiratory burst amplitude. Top: representative traces recorded during sigh refractory period show effects of bilateral preBötC long pulse photostimulation (LPP) of preBötC GRPR (brown) or NMBR (blue) neurons on eupneic breathing f and $V_T$. Bottom: $V_T$ and f of eupneic breaths and during photostimulation. (paired t-tests, $GrprF^{lp}$ +AAV-ChR2, n=4 mice, f: $t_3$=−8.13801, p=4 × 10⁻³; $V_T$: $t_3$=−5.035, p=0.015; $Nmbr$-ChR2, n=5 mice, f: $t_4$=7.33379, p=2 × 10⁻³, $V_T$: $t_4$=−5.875, p=4 × 10⁻³). Normalized (norm) $V_T$ and f traces relative to stimulus onset measured prestimulation, during stimulation and poststimulation. (**b**) Distinct effects of activating preBötC GRPR-only (brown) and NMBR-only (blue) neurons on inspiratory burst amplitude. Top and bottom (left): representative traces show

*Figure 6 continued on next page*

*Figure 6 continued*

the effects of bilateral preBötC SPP and LPP of preBötC GRPR-only (brown) or NMBR-only (blue) neurons on breathing $V_T$. Bottom (right): $V_T$ of eupneic breaths and during SPP stimulation. Increase in $V_T$ in response to bilateral preBötC SPP of preBötC GRPR-only (brown) or NMBR-only (blue) neurons (paired t-tests, GRPR-only, n=4 mice: $t_3$=0.746, p=0.510; NMBR-only, n=4 mice: $t_3$=–6.886, p=6 × $10^{-3}$). (**c**) Activation of hM3Dq receptors expressed on NMBR neurons with clozapine-n-oxide (CNO) significantly increased breathing amplitude and decreased $f$ (paired two-tailed t-tests, n=3 mice, $f$: $t_2$=5.71, p=0.03; $V_T$: $t_2$=5.47, p=0.03). (**d**) Sighs can be induced after GRPR and NMBR inhibition. Left: schematic depicting bilateral placement of optical cannula targeting parafacial (pF) or preBötC. Middle: representative traces showing LPP of pF GRP neurons, or preBötC GRPR and NMBR neurons elicited ectopic sighs (doublet, red arrowhead), but not of pF NMB neurons, in the presence of GRPR antagonist RC3095 and NMBR antagonist BIM23042. Right: $V_T$ of doublets induced by pF LPP of GRP neurons was not significantly different from that of endogenous sighs (n=4 mice; $t_3$=0.724, p=0.522); $T_I$ increased compared to endogenous sighs ($t_3$=–7.493, p=5 × $10^{-3}$), $T_E$ of the following respiratory cycle was unaffected ($t_3$=–0.472, p=0.669). $V_T$ and $T_I$ of the doublets elicited by preBötC LPP of GRPR (n=5 mice, $V_T$: $t_4$=0.199, p=0.852; $T_I$: $t_4$=0.912, p=0.413) or NMBR (n=4 mice, $V_T$: $t_3$=–0.845, p=0.460; $T_I$: $t_3$=–0.376, p=0.732) neurons after blockade of both receptors were not different from spontaneous sighs. $T_E$ immediately following doublet increased in $Grpr^{Flp}$ (n=5 mice, $t_4$=–3.144, p=0.035), but not lengthened in $Nmbr$-ChR2 mice (n=4 mice, $t_3$=–0.801, p=0.482). Data are shown as mean ± SE. Asterisks indicate post-hoc multiple comparison test or paired t-test results: *, significance with p<0.05.

The online version of this article includes the following figure supplement(s) for figure 6:

**Figure supplement 1.** GRP and NMB receptors in the preBötzinger Complex (preBötC) effectively blocked by RC3095 + BIM23042 microinjections.

GRPR or NMBR neurons still elicited sighs of doublet shape. $V_T$ and $T_I$ of the sighs elicited in $Grpr^{Flp}$ or $Nmbr$-ChR2 mice after blockade of both receptors was no different from spontaneous sighs (**Figure 6d**). Postsigh apnea was observed in $Grpr^{Flp}$ ($T_E$ immediately following the sigh increased to 132 ± 24%), but not in $Nmbr$-ChR2 mice (**Figure 6d**).

Thus, initiating or increasing the activity of pF GRP neurons, or preBötC GRPR or NMBR neurons, can generate sighs without activation of these receptors, either endogenously or exogenously.

## preBötC SST neurons mediate sighing

Can activity of preBötC neurons other than those expressing $Grpr$ or $Nmbr$ generate sighs? To address this question, we virally expressed ChR2, hM3Dq, or the PSAM4 (**Figure 7—figure supplement 1**) in preBötC SST neurons. In ChR2-transduced $Sst^{Cre}$ mice, bilateral preBötC SPP during inspiration elicited ectopic sighs of partial doublets or augmented breath shapes and reset the sigh cycle (**Figure 7b**). $V_T$, $T_I$, and $T_E$ of the induced sighs were no different from those of spontaneous sighs (**Figure 7c**). The ectopic sighs elicited by photostimulation of preBötC SST neurons exhibited a post-sigh apnea, $T_E$ of the following respiratory cycle increased by 210 ± 19% (**Figure 7c**). After BIM23042 and RC3095 microinjection (300 µM each, 50 nl/side), spontaneous sighs were eliminated (**Figure 6—figure supplement 1**). In the presence of antagonists, preBötC SPP during inspiration could still elicit an ectopic sigh (**Figure 7d**). $V_T$ and $T_I$ of ectopic sighs elicited after blockade of both receptors were no different from spontaneous sighs, or from ectopic sighs elicited before blockade (**Figure 7c**). Ectopic sighs elicited after blockade also had a postsigh apnea, $T_E$ of the following respiratory cycle increased by 236 ± 49% (**Figure 7c**).

In hM3Dq-transduced $Sst^{Cre}$ anesthetized mice (**Figure 7e**), excitation of preBötC SST neurons by CNO application onto ventral brainstem surface affected eupneic breathing: $f$ decreased by 60 ± 6.2%, $V_T$ increased by 222 ± 21%. CNO application increased sigh rate ~10 fold with sighs appearing as doublets (**Figure 7f**). In these mice, subsequent microinjection of BIM23042 and RC3095 into preBötC did not affect $f$, $V_T$, or sigh rate induced by CNO administration (**Figure 7f**). Thus, excitation of preBötC SST neurons can produce sighs in an NMBR- and GRPR-independent manner.

Next, we investigated whether sighs induced by the NMBR- and GRPR-dependent pathway require activity of preBötC SST neurons. For that, we transfected preBötC SST neurons with Cre-dependent ultrapotent inhibitory DREADD AAV-PSAM4-GlyR *Magnus et al., 2019* in $Sst^{Cre}$ mice (**Figure 7g**). The sigh rate was then increased by microinjection of peptides NMB and GRP (250 µM each, 50 nl/side) into preBötC (**Figure 7g and h**) in ketamine/xylazine anesthetized mice, which also decreased $f$, and increased $V_T$ (**Figure 7g and h**). Subsequent application of the PSAM4-GlyR ligand uPSEM817 to the brainstem surface did not affect $f$ or $V_T$, but completely eliminated all sighs (**Figure 7g and h**). See **Figure 7—figure supplement 2** for control applications of uPSEM817 and saline. Note that similar to excitatory DREADD effects in $Nmbr^{Cre}$ mice, manipulations increasing $V_T$ of all breaths were classified to affect $V_T$ of eupneic breaths rather than induce an 'all sigh' rhythm, according to our definition of sighs (**Appendix 1—figure 1**).

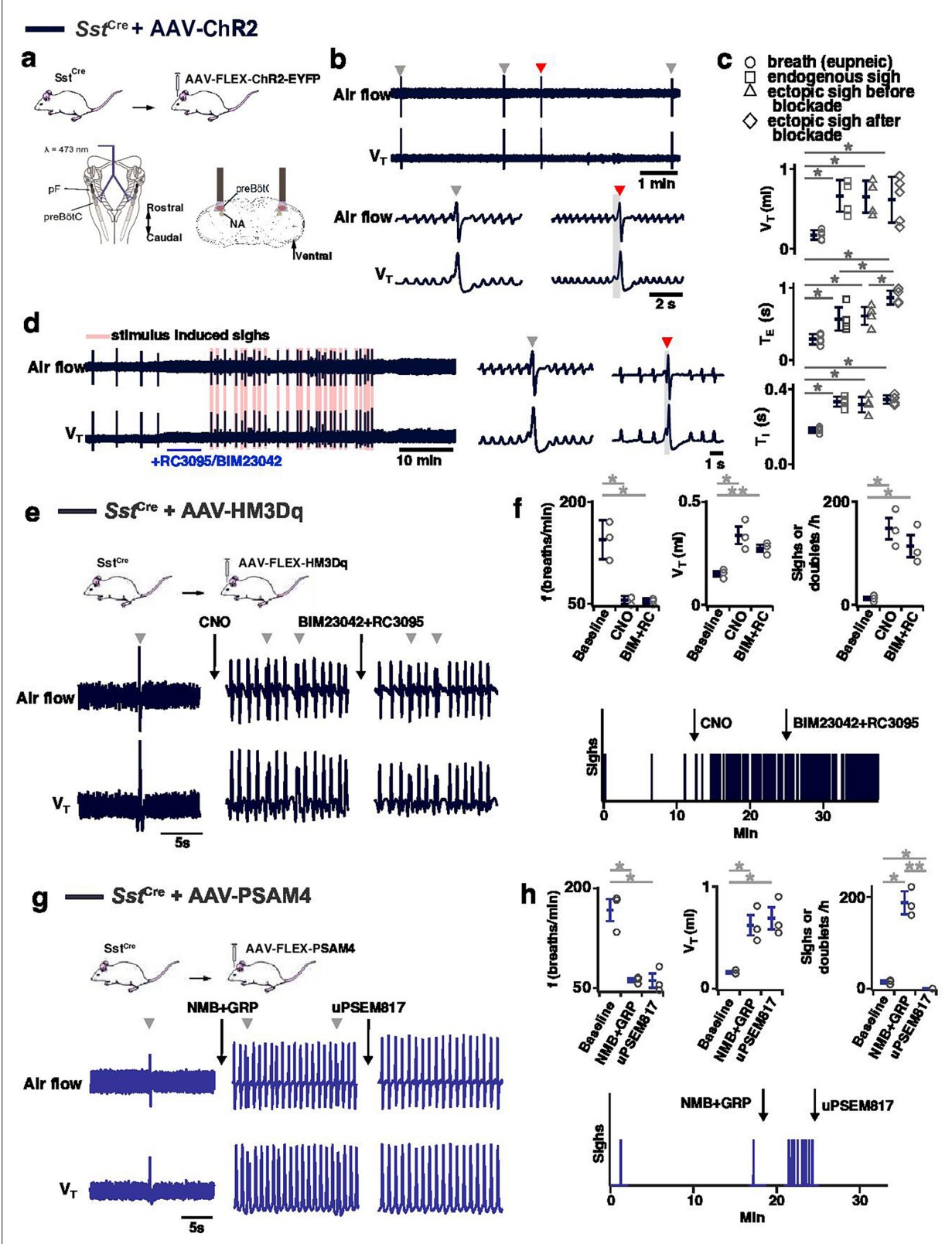

**Figure 7.** Effects of activation or inhibition of preBötzinger Complex (preBötC) SST neurons on sigh generation. (**a–d**) Optogenetic activation of SST neurons generates sighs after blockade of NMBRs and GRPRs. (**a**) Top: Schematic of the genetic strategy to target preBötC SST neurons. Bottom: Schematic depicting bilateral placement of optical cannula targeting preBötC. (**b**) Ectopic sigh (red arrowhead) elicited by bilateral SPP (gray box) of preBötC SST neurons. (**c**) $V_T$ ($t_4$=0.176, p=0.863), $T_I$ ($t_4$=1.355, p=0.200), $T_E$ ($t_4$=0.734, p=0.477) of ectopic sighs before blockade or $V_T$ ($t_4$=0.856, p=0.409),

*Figure 7 continued on next page*

*Figure 7 continued*

$T_I$ ($t_4$=0.894, p=0.389) of ectopic sighs after blockade of NMBR and GRPR were no different from endogenous sighs (n=5 mice). $T_E$ of ectopic sighs elicited after blockade was longer than that of endogenous sighs ($t_4$=4.960, p=3 × 10$^{-4}$). $V_T$, $T_I$, and $T_E$ of ectopic sighs before ($V_T$: $t_4$=7.925, p=4 × 10$^{-6}$; $T_I$: $t_4$=11.194, p=10$^{-7}$; $T_E$: $t_4$=5.405, p=10$^{-4}$) or after ($V_T$: $t_4$=7.245, p=10$^{-5}$; $T_I$: $t_4$=13.442, p=10$^{-8}$; $T_E$: $t_4$=9.631, p=5 × 10$^{-7}$) blockade were increased compared with eupneic breaths (n=5 mice), indicative of augmented breaths with postsigh apneas. Statistical significance was determined with a One-Way RM ANOVA followed by All Pairwise Multiple Comparison Procedures (Holm-Sidak method), $V_T$: $F_{3,16}$ = 30.357, p<0.001; $T_I$: $F_{3,16}$ = 78.526, p<0.001; $T_E$: $F_{3,16}$ = 31.134, p<0.001. (**d**) SPP elicits sighs in the presence of GRPR antagonist RC3095 and NMBR antagonist BIM23042. (**e, f**) Chemogenetic activation of SST neurons generated sighs after blockade of NMBRs and GRPRs. (**e**) Top: schematic diagram of genetic strategy to selectively express DREADD receptor hM3Dq on preBötC SST neurons. Bottom: representative trace of airflow and $V_T$ during baseline, after application of clozapine-n-oxide (CNO), and after microinjection of RC3095 and BIM23042. (**f**) Top: Activation of hM3Dq receptors expressed on preBötC SST$^+$ neurons significantly decreased breathing *f*, increased $V_T$, and elevated sigh frequency; subsequent BIM23042 and RC3095 (B+R) microinjection into preBötC did not significantly affect breathing *f*, $V_T$, nor sigh rate induced by CNO application (repeated measures ANOVA, n=3 mice; *f*: $F_{2,4}$ = 28.3, p=0.004; $V_T$: $F_{2,4}$ = 25.7, p=0.005; sigh rate: $F_{2,4}$ = 26.1, p=0.005). Bottom: representative trace depicting sighs during baseline, after application of CNO, and after microinjection of NMBR and GRPR antagonists RC3095 and BIM23042. (**g**) Top: schematic diagram of genetic strategy to selectively express ultrapotent inhibitory DREADD receptor PSAM4-GlyR on preBötC SST$^+$ neurons. Bottom: representative trace of airflow and $V_T$ during baseline, after preBötC microinjection of peptides NMB and GRP (250 µM each, 50 nl/side), and application of uPSEM817 (10 mM, 30 µl applied to brainstem surface). (**h**) Top: Microinjection of peptides NMB and GRP into preBötC significantly decreased breathing *f*, increased $V_T$, and elevated sigh frequency; subsequent inhibition of SST + preBötC neurons selectively eliminated any sighs, but preserves decreased *f* and $V_T$ (repeated measures ANOVA, n=3 mice; *f*: $F_{2,4}$ = 55.8, p=0.001; $V_T$: $F_{2,4}$ = 19.0, p=0.009; sigh rate: $F_{2,4}$ = 83.8, p<0.001). Bottom: representative trace depicting sighs during baseline, after bilateral preBötC microinjection of NMB and GRP, and after application of PSAM4-GlyR ligand uPSEM817 to the brainstem surface. Data are shown as mean ± SE. Asterisks indicate post-hoc multiple comparison test or paired t-test results: *, significance with p<0.05; **, significance with p<0.01.

The online version of this article includes the following figure supplement(s) for figure 7:

**Figure supplement 1.** Immunohistochemical verification of ChR2, HM3Dq and PSAM4 expression in preBötC SST neurons.

**Figure supplement 2.** Control applications of uPSEM817 or saline.

# Discussion

In all mammals, sighs are generated periodically to maintain lung function and, in humans, are episodically associated with such emotional states as relief, sadness, yearning, exhaustion, stress, and joy (***Knowlton and Larrabee, 1946***; ***Mccutcheon, 1953***; ***Reynolds, 1962***; ***Ramirez, 2014***; ***Li and Yackle, 2017***), and in rodents are associated with inferred indicators of emotional states such as stress and anxiety (***Carnevali et al., 2013***). The signals producing these large periodic inspiratory efforts originate within the preBötC, a heterogeneous medullary population that is the kernel for generation of breathing rhythm (***Del Negro et al., 2018***). Molecularly defined preBötC subpopulations underlie different functions during breathing (***Del Negro et al., 2018***). Ablation, genetic deletion, or pharmacological excitation or inhibition of preBötC neurons expressing receptors (NMBR and GRPR) for bombesin-related peptides (NMB and GRP) significantly affects rhythmic (~25–40/hr) sighing in rodents (***Li et al., 2016***). Here, we further explored the mechanisms of sigh generation.

Sighs exhibit various shapes, i.e., augmented single inspiratory effort, partial doublets, and full doublets, that have two unifying characteristics: (i) significant transient increase in $V_T$ and; (ii) occur at a much lower frequency compared to eupneic breaths; often these augmented inspiratory efforts are followed by a lengthened interburst interval, i.e., postsigh apnea. We use these criteria as a definition of sighs, which unifies observations and conclusions in this and previous in vivo ***Ramirez, 2014***; ***Carnevali et al., 2013***; ***Kam et al., 2013***; ***Sherman et al., 2015*** and in vitro (***Lieske et al., 2000***; ***Feldman and Gray, 2000***) studies that used various, often incongruent definitions of sighs.

Here, we investigated two novel properties of sigh generation: sigh phase and postsigh refractory period. By using the same approach that we employed for differentiating the effects of optogenetic stimulation on various preBötC subpopulations on the eupneic cycle (***Cui et al., 2016***; ***Sherman et al., 2015***; ***de Sousa Abreu et al., 2022***), we found that sighs evoked by optogenetic stimulation of various subpopulations reset the sigh cycle and delayed the onset of the next endogenous sigh. Also, similar to a postinspiratory refractory period in the respiratory cycle, during which ectopic breaths cannot be evoked immediately following termination of an inspiratory effort (***Cui et al., 2016***), we similarly observed a postsigh refractory period following an endogenous sigh, during which ectopic sighs could not be evoked. We suggest that these subpopulations have intrinsic 'sigh-generating' properties, e.g., unique signaling pathways or connectivity that affects excitability of these subpopulations. Although vagotomy transiently abolishes

eupneic sighs (*Cherniack et al., 1981*), they can be evoked in the absence of sensory feedback in vivo after vagotomy *Janczewski et al., 2012* or in vitro (*Lieske et al., 2000*; *Feldman and Gray, 2000*), suggesting that sensory inputs by themselves are not necessary, but likely modulate activity of these subpopulations to affect frequency of sighs. The mechanisms that underlie the slow sigh rhythm (~minutes) remains to be discovered, and might involve intracellular $Ca^{2+}$ oscillations (*Borrus et al., 2024*).

In mice, sigh-like movements appear in utero between E17 and E18 (*Chapuis et al., 2014*), when at least two types of inspiratory-modulated neurons can be differentiated in rhythmic slices: neurons that fire in sync with regular 'eupneic' XII bursts, and those that fire only during sigh-like bursts (*Chapuis et al., 2014*). This could result from increased activity in a single population with a distribution of activation thresholds or from recruitment of a distinct high-threshold population. In vitro, the majority of NMBR preBötC neurons recorded were active in both eupneic and sigh bursts, a few GRPR preBötC neurons recorded were active in eupnea, but none were active exclusively during sighs, q.v., (*Chapuis et al., 2014*). Rhythmic preBötC NMBR neurons in neonatal mice were active during both eupneic and sigh bursts (*Figure 4b*), but not during burstlets or the pre-I period. These results suggest that NMBR preBötC neurons are Type II neurons that participate in generation of the burst patterns of sighs and eupneic breaths, but are not rhythmogenic.

Sighs can be induced by activation of various subpopulations within two structures: NMB or GRP in pF and NMBR, GRPR, or SST in preBötC. We previously reported a microcircuit capable of generating sighs involving pF neurons producing bombesin-related peptides that project to preBötC neurons expressing cognate NMBRs and GRPRs (*Li et al., 2016*), and that activation of these preBötC neurons via their peptide receptors was sufficient to generate sighs (*Li et al., 2016*). Here, in significantly extending this work, we show that sighs can be evoked by direct photoexcitation of GRP, NMBR, or GRPR neurons, bypassing activation of these receptors. Photostimulation of preBötC GRPR or NMBR neurons can elicit ectopic sighs (*Figure 3a and c*), even in the presence of GRPR and NMBR antagonists at sufficient concentration to block the effect of exogenous application of peptides that increase sigh rate (*Figure 6d*, *Figure 6—figure supplement 1*). Thus, sighs can be induced by increasing the excitability of these subpopulations, without obligatory participation of pathways activated by bombesin-related peptides. In other words, while it can provide sufficient external input to trigger sighs, activation of preBötC NMBRs and/or GRPRs is not necessary for sigh production. This suggests that sighs are not the unique product of a preBötC bombesin-peptide signaling pathway.

We studied photoactivation of pF *Nmb-* and *Grp*-expressing neurons in transgenic mice. Could our protocol have also activated GRP or NMB-containing presynaptic terminals originating from brainstem neurons outside pF, such as found in the parabrachial nuclei and NTS (*Li et al., 2016*)? This is unlikely in the case of NMB, since *Nmb*-expressing neurons are limited to pF in the medulla (*Li et al., 2016*). Also, unlikely the case of GRP, as there is no evidence that pF neurons express GRPRs (*Kamichi et al., 2005*). However, our protocol would result in the expression of ChR2 in neurons that expressed *Nmb-* or *Grp*-expressing neurons at any point since conception, even transiently. Thus, we cannot rule out that; in our adult mice there were neurons in pF that once expressed these peptides but no longer did so, and that these vestigial NMB or GRP neurons could have been perturbed in our protocol. Such determination requires additional studies outside the scope of the work reported here.

preBötC neurons expressing *Sst*, mostly distinct from *Nmbr* and *Grpr* populations (*Figure 2*), receive input from rhythmogenic preBötC neurons, many have an inspiratory firing pattern (*Ashhad and Feldman, 2020*), that appear to serve a premotor role to shape inspiratory motor output (*Cui et al., 2016*). Photostimulation of preBötC SST neurons increases peak inspiratory amplitude in both anesthetized and awake mice (*Cui et al., 2016*; *de Sousa Abreu et al., 2022*). Here, DREADD-mediated excitation of SST neurons significantly increased sigh rate, with the effect much larger than DREADD-mediated excitation of NMBR neurons (16- vs threefold increase). Additionally, unlike the significant 1–5 s latency to induce an ectopic sigh when photostimulating GRPR or NMBR preBötC neurons (*Figure 3e*), photostimulating preBötC SST neurons during inspiration induced a second augmented inspiration immediately following the eupneic burst to convert it into a sigh (*Figure 7b*). Importantly, the effects of such photostimulation were unaffected by blockade of both NMBR and GRPR receptors (*Figure 7d*), and sighs evoked by microinjection of peptides NMB and GRP into preBötC were completely eliminated by inhibition of SST neurons that expressed an inhibitory DREADD PSAM4 (*Figure 7g and h*). We conclude that activation of preBötC SST neurons is also

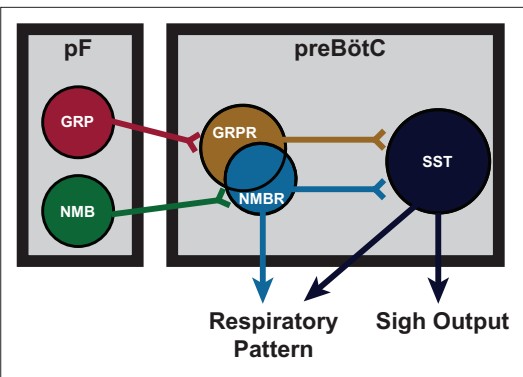

**Figure 8.** Proposed model of sigh generating parafacial (pF)-preBötzinger Complex (preBötC) microcircuit. NMB and GRP neurons in pF project to preBötC neurons expressing cognate receptors NMBR and GRPR. NMBR and GRPR preBötC neurons mediate sigh output via connections to downstream SST preBötC neurons, but also directly contribute to the pattern of eupneic breathing.

sufficient to induce sighs and further suggest they can induce sighs, downstream from NMBR and GRPR preBötC neurons, and may be necessary to do so (*Figure 8*).

Our initial delineation of a microcircuit of four subpopulations, i.e., pF NMB and GRP neurons and preBötC NMBR and GRPR neurons, that affected sighs led us to hypothesize its role in sigh generation, but we were agnostic as to its role in the production of eupneic breaths (*Li et al., 2016*). Our data here, consistent with a previous report of effects from intravenous administration of bombesin (*Kaczyńska and Szereda-Przestaszewska, 2009*), suggests that this microcircuit is tightly integrated within the eupneic CPG with NMBR, GRPR, and SST populations having distinct though overlapping roles in modulation of the eupneic breathing pattern as well as in the generation of sighs. The majority of preBötC neurons expressing NMBR, GRPR (or both) receptors are glutamatergic and excitatory (*Figure 2*). In vitro, no NMBR preBötC neurons were active during burstlets or the pre-I period, suggesting that NMBR preBötC neurons are Type II neurons that participate in generation of breathing pattern but not rhythm generation (similar to the properties of preBötC SST neurons *Ashhad and Feldman, 2020*). In vivo, we observed distinctly different effects from activation of preBötC GRPR and NMBR neurons: photoactivation of GRPR neurons increased breathing frequency, whereas photo- or chemo-activation of NMBR neurons decreased breathing frequency and increased amplitude (*Figure 6a and c*); otherwise, their effects on sighing appeared similar (*Figure 3*). Notably, some neurons express both NMBR and GRPR receptors (*Figure 2*). When the overlapping subpopulation of preBötC neurons expressing both GRPRs and NMBRs was excluded, we could activate the two discrete pathways independently. Photostimulating GRPR-only neurons had no effect on inspiratory burst amplitude, whereas photostimulating NMBR-only neurons increased amplitude (*Figure 6b*). Additionally, DREADD-mediated excitation of the SST population increased both the sigh rate and the inspiratory burst amplitude (*Figure 7f*). Microinjection of peptides NMB and GRP into preBötC increased sigh rate and inspiratory burst amplitude; subsequent ultrapotent chemogenetic inhibition of preBötC SST neurons selectively abolished sighs, but not an increase in inspiratory amplitude (*Figure 7g and h*), suggesting that modulation of breathing pattern by NMBR and GRPR neurons is independent of SST neurons (*Figure 8*).

Sighing is a relatively straightforward behavior that appears to result from a delineable microcircuit that could only be discovered because of the close relationship between the essential, readily localizable, compact circuitry with a definitive precisely measurable functional physiologically relevant output. How this unexpected complexity for sigh generation reflects on strategies for unraveling ever more complex and difficult to quantify behaviors remains to be determined.

## Methods
Experimental procedures were carried out in accordance with the United States Public Health Service and Institute for Laboratory Animal Research Guide for the Care and Use of Laboratory Animals. All animals were handled according to institutional protocols at the University of California, Los Angeles (#1994-159-83P0) and approved by the University of California Animal Research Committee (Animal Welfare Assurance #A3196-01). Every effort was made to minimize pain and discomfort, as well as the number of animals.

## Animals

*Grp*^Cre was obtained from Jackson Labs (Strain #033174). *Grpr*^flp, *Nmb*^Cre, and *Nmbr*^Cre animals were designed and generated by Cyagen Biosciences Inc (see below). Transgenic ChR2 mice were generated by crossing Cre mice with floxed-ChR2-tdTomato mice (Jackson Labs Strain #012567). These crosses generated mice expressing ChR2 in GRP neurons (*Grp*-ChR2), NMB neurons (*Nmb*-ChR2), or NMBR neurons (*Nmbr*-ChR2). To express ChR2 selectively in preBötC neurons, AAV injections were performed on adult *Sst*^Cre mice (Jackson Labs Strain #013044) or *Grpr*^flp mice. All in vivo experiments were performed on adult male or female mice (10–14 wk old, 24–32 g). Unless otherwise specified, mice were anesthetized with ketamine/xylazine. For in vitro experiments, transgenic *Grpr*-EGFP, *Nmbr*-tdTomato, and *Nmbr*-GCamp6s were used. These mice were generated by crosses between *Nmbr*^Cre or *Grpr*^Flp mice and one of the following lines: Ai14 (Jackson Labs Strain #007914), RCE:FRT (MMRRC Strain #032038-JAX), GCaMp6f (Jackson Labs Strain #029626).

To generate *Nmb*^Cre mice the open reading frame of Cre recombinase together with SV40 polyA signal was cloned downstream of the mouse *Nmb* promoter, such that the expression of Cre mimicked the endogenous mouse *Nmb* gene. The PiggyBac ITRs was inserted into the BAC backbone flanking the genomic insert, to facilitate transposes mediated BAC integration. The modified BAC was co-injected with transposes into single-cell stage fertilized eggs from C57BL/6 mice. Resulting pups were then genotyped by PCR for the presence of the modified BAC and the Cre cassette. To generate *Grpr*^Flp mice, the *Grpr* gene was first located on mouse chromosome X. Three exons have been identified, with the ATG start codon in exon 1 and TAG stop codon in exon 3; the TAG stop codon was then replaced with the Flp cassette. To engineer the targeting vector, homology arms were generated by PCR using BAC clone RP23-402H23 and RP23-50M18 from the C57BL/6 J library as template. In the targeting vector, the Neo cassette was flanked by FRT sites. DTA was used for negative selection. The constitutive knock-in allele was obtained after FRT-mediated recombination. C57BL/6 ES cells were then used for gene targeting to introduce knock-in alleles into host embryos followed by transfer into surrogate mothers. Resulting pups were then genotyped by PCR for the presence of the Flp cassette. To generate *Nmbr*^Cre mice, the *Nmbr* gene was first located on mouse chromosome 10. Three exons have been identified, with the ATG start codon in exon 1 and TGA stop codon in exon 3; the TGA stop codon was then replaced with the Cre cassette. To engineer the targeting vector, homology arms were generated by PCR using BAC clone RP24-232I16 and RP24-124K10 from the C57BL/6 J library as template. In the targeting vector, the Neo cassette was flanked by LoxP sites. DTA was used for negative selection. The constitutive knock-in allele was obtained after Cre-mediated recombination. C57BL/6 ES cells were then used for gene targeting to introduce knock-in alleles into host embryos followed by transfer into surrogate mothers. Resulting pups were then genotyped by PCR for the presence of the Cre cassette.

## Viral vector design

pAAV-hSyn Con/Foff hChR2(H134R)-EYFP (Addgene plasmid # 55646; http://n2t.net/addgene:55646; RRID:Addgene_55646) and pAAV-hSyn Coff/Fon hChR2(H134R)-EYFP (Addgene plasmid # 55648; http://n2t.net/addgene:55648; RRID:Addgene_55648) were gifts from Karl Deisseroth (*Fenno et al., 2020*). pAAV-hSyn-DIO-hM3D(Gq)-mCherry was a gift from Bryan Roth (Addgene plasmid # 44361; http://n2t.net/addgene:44361; RRID:Addgene_44361). pAAV-Syn-flex-PSAM4-GlyR-IRES-EGFP was a gift from Scott Sternson (*Magnus et al., 2019*) (Addgene plasmid # 119741; http://n2t.net/addgene:119741; RRID:Addgene_119741). All viruses were stored in aliquots at –80 °C until use.

## Viral injections

Mice were anesthetized with isoflurane (4% for induction and 2% for maintenance) and placed in a stereotaxic apparatus (David Kopf Instruments) with Bregma and Lambda skull landmarks level. Two holes were drilled in the skull 6.80 mm caudal to Bregma and 1.2 mm lateral to the midline. Virus was delivered 4.65 mm from the dorsal surface of the brain into the preBötC through a glass pipette using either a pressure ejection system (Picospritzer II; Parker Hannafin) or Micro4 microinjection system (World Precision Instruments). For ChR2 experiments, we used 100–200 nl of virus solution (1–6 × $10^{12}$ vg/ml) per side, and for DREADD experiments, 30–50 nl of virus solution (4.6 × $10^{12}$ vg/ml) per side. The pipettes were left in place for 5 min after injection to minimize backflow. The wound was closed with 5–0 gauge non-absorbable sutures. Mice were returned to their home cage and given 2–3 wk to

recover to allow for sufficient levels of protein expression. All virus injection sites were subsequently confirmed by immunostaining and only mice with sufficient and localized transfection in preBötC were used for all analysis.

## Surgical procedures for ventral approach

Adult mice were anesthetized via intraperitoneal injection of ketamine and xylazine (100 and 10 mg/kg, respectively). Isoflurane (1–2% volume in air) was administered throughout an experiment. The level of anesthesia was assessed by the suppression of the withdrawal reflex. A tracheostomy tube was placed in the trachea through the larynx, and respiratory flow was detected with a flow head connected to a transducer to measure airflow (Mouser Electronics). The mice were placed in a supine position in a stereotaxic instrument (David Kopf Instruments). The larynx was denervated, separated from the pharynx, and moved aside. The basal aspect of the occipital bone was removed to expose the ventral aspect of the medulla. The canal of the hypoglossal nerve (XII) served as a suitable landmark. The preBötC was 0.15 mm caudal to the hypoglossal canal, 1.2 mm lateral to the midline, and 0.24 mm dorsal to the ventral medullary surface.

## Photostimulation

A 473 nm laser (OptoDuet Laser; IkeCool) was targeted bilaterally to the preBötC or pF with a branching fiber patch cord (200 µm diameter; Doric Lenses) brought to the exposed ventral medullary surface. Laser power was set at 5 mW. Short Pulse Photostimulation (SPP; 200 ms) and Long Pulse Photostimulation (LPP; 5 s) waveforms were delivered under the command of a pulse generator (Pulsemaster A300 Generator; WPI) connected to the laser power supply. In instances where breathing frequency was decreased, e.g., after BIM23042 and RC3095 microinjections, SPP and LPP of longer durations (500 ms and 8 s, respectively) were used to confirm consistency of results. Stimulating 500 µm rostral to preBötC or pF in mice expressing ChR2 that served as a control did not produce significant output effects (*Figure 1—figure supplement 2* & *Figure 3—figure supplement 3*). No output effects and no sighs were produced by preBötC or pF photostimulation in *Grp*-RFP, *Grpr*-EGFP, *Nmbr*-EGFP, *Nmb*-RFP reporter mice (*Figure 1—figure supplement 2* & *Figure 3—figure supplement 3*).

## Pharmacological injection experiments

The NMBR antagonist BIM23042 (Tocris Bioscience) and the GRPR antagonist RC3095 (Sigma-Aldrich) were injected together (300 µM each, 50–60 nl/side) to block NMBR and GRPR activation. Injections were made using micropipettes (~40 µm tip), placed bilaterally into the preBötC. Injections targeted to the center of the preBötC were placed 0.15 mm caudal to the hypoglossal canal, 1.2 mm lateral to the midline, and 0.24 mm dorsal to the ventral medullary surface (*Cui et al., 2016*). Small corrections were made to avoid puncturing of blood vessels on the surface of the medulla. All injections were made using a series of pressure pulses (Picospritzer, Parker-Hannifin; or Micro4, World Precision Instruments). For activation of hM3Dq receptors, 50 µL of CNO solution was applied to the ventral surface of the brainstem. CNO (Tocris Biosciences) was dissolved in saline at a concentration of 1 mg/ml with 0.5% dimethylsulfoxide. Solution was prepared fresh daily. For activation of PSAM4-GlyR receptors, 30 µL of uPSEM817 tartrate solution (10 mM) was applied to the ventral surface of the brainstem. In chemogenetic experiments CNO or uPSEM817 were applied to the ventral surface of the brainstem instead of microinjection in order to avoid preBötC damage from multiple microinjections; in control experiments application of the same dose of CNO or uPSEM817 does not produce responses in any breathing parameters.

## In situ hybridization and immunostaining

*Nmbr*^Cre, *Sst*^Cre, or wild-type mice (10–14 wk) were euthanized with isoflurane overdose, their brainstems were rapidly removed and flash-frozen in dry ice. Fresh frozen brainstems were sectioned sagittally on a cryostat (CryoStar NX70, Thermo Scientific), mounted on SuperFrost Plus slides (Fisher Scientific) and stored at –80 °C for at least 1 d. They were then processed according to the manufacturer's protocol (RNAscope version 1, Advanced Cell Diagnostics). Briefly, tissue samples were postfixed in 10% neutral buffered paraformaldehyde, washed, and dehydrated in sequential concentrations of ethanol (50, 70, and 100%). Samples were treated with protease IV and incubated for 2 hr at 40 °C in the HybEZ Hybridization Oven (Advanced Cell Diagnostics) in the presence of target probes. We used

combinations of the following probes for hybridization: *Mm-ChAT, Mm-Grpr, Mm-Nmbr, Mm-Sst, Mm-Slc17a6 (VGluT2), Mm-Aldh1l1, Cre,* and *Flp.* A probe for *Mm-ChAT* was used in all probe combinations to mark the location of nucleus ambiguus, which is necessary for determining the location of preBötC. After a 4-step amplification process, samples were counterstained with DAPI and coverslipped with ProLong Gold (Invitrogen) used as a mounting agent. Images were acquired on a confocal laser scanning microscope (LSM710 META, Zeiss). High-resolution z-stack confocal images were taken at 1 µm intervals and then merged using ImageJ software.

Immunohistochemistry was performed according to the following protocol. Free-floating sections were rinsed in PBS and incubated with 10% normal donkey antiserum (NDS) and 0.2% Triton X-100 in PBS for 60 min to reduce nonspecific staining and increase antibody penetration. Sections were incubated overnight with primary antibodies diluted in PBS containing 1% NDS and 0.2% Triton X-100. The following day, sections were washed in PBS, incubated with the specific secondary antibodies conjugated to the fluorescent probes diluted in PBS for 2 hr. Sections were further washed in PBS, mounted, and coverslipped with Fluorsave mounting medium (Millipore). The primary antibodies used for this study were as follows: rabbit polyclonal anti-somatostatin-14 (1:500; Peninsula Laboratories), mouse monoclonal anti-NeuN (1:500; Millipore; MAB377), goat anti-ChaT (1:500, Chemicon), and chicken polyclonal anti-GFP (1:500; Aves Labs). DyLight488 donkey anti-chicken, Rhodamine Red-X donkey anti-rabbit, and Cy5 donkey anti-mouse conjugated secondary antibodies (1:250; Jackson ImmunoResearch) were used to detect primary antibodies. Slides were observed under an AxioCam2 Zeiss fluorescent microscope connected with AxioVision acquisition software or under a LSM510 Zeiss confocal microscope with Zen software (Carl Zeiss). Images were acquired, exported in TIFF files, and arranged to prepare final figures in Zen software (Carl Zeiss) and Adobe Photoshop (Adobe).

## In vitro electrophysiology

Neonatal mice (P0-5) of either sex were anesthetized with isoflurane. The brainstem was isolated from the pons to the rostral cervical spinal cord under cold ACSF containing (in mM): 124 NaCl, 3 KCl, 1.5 $CaCl_2$, 1 $MgSO_4$, 25 $NaHCO_3$, 0.5 $NaH_2PO_4$, and 30 D-glucose; a single transverse slice (600 µm) was cut on a vibratome (Leica VT1200S).

Slices were placed in a recording chamber with the rostral face of preBötC facing up and continuously superfused with warmed ACSF (at 28–30°C) equilibrated with 95% $O_2$ and 5% $CO_2$, and extracellular $K^+$ was increased to 9 mM to elevate preBötC excitability. Inspiratory-related motor output was recorded from the hypoglossal nerve (XII) using a suction electrode and a differential AC amplifier (AM systems), filtered at 2–4 kHz, integrated, and digitized at 10 kHz. Whole-cell recordings were made using a MultiClamp 700 A (Molecular Devices), filtered at 2–4 kHz, and digitized at 10 kHz. Borosilicate glass recording electrodes (G150-4, Warner Instruments) had tip resistances of 4–8 MΩ and were filled with intracellular solution containing (in mM): 140 potassium gluconate, 10 HEPES, 5 NaCl, 1.1 EGTA, 2 Mg-ATP, and 0.1 $CaCl_2$ (~305 mOsm; pH 7.3). In two-photon imaging experiments, field potentials were recorded from the contralateral preBötC, using an extracellular electrode and a differential AC amplifier (AM systems), filtered at 2–4 kHz, integrated, and digitized at 10 kHz. Digitized data were analyzed offline using custom procedures written for IgorPro (Wavemetrics). Agonists were bath applied at the specified concentrations while monitoring field potentials and XII output.

## Two-photon calcium imaging

Intracellular $Ca^{2+}$ was imaged using a multi-photon resonant scanner (3i) and a microscope equipped with a water immersion 20 x, 1.0 objective. Illumination was provided by a laser with a power output of 1050 mW at 970 nm (Coherent Chameleon Ultral). Identified preBötC neurons were scanned at 10–39 Hz and fluorescence data were collected using 3i software and analyzed using SlideBook, Image J, and IgorPro. Regions of interest (ROIs) were manually detected and $Ca^{2+}$ transients were plotted as $F/F_0$, where $F_0$ is the average fluorescence intensity of all pixels within a given ROI averaged over the entire time series.

## Statistical analysis

In vitro data were analyzed using IgorPro (Wavemetrics) and GraphPad Prism (GraphPad Software) was used for statistical analysis. Paired T-tests or Kruskal-Wallis one-way ANOVA, adjusted for multiple comparisons, was used to calculate significance.

For in vivo experiments, data were recorded on a computer using LabChart 7 Pro (AD Instruments) and analyzed using LabChart 7 Pro (ADInstruments), Excel, and Igor Pro (Wavemetrics, Inc) software. The flow signal was high-pass filtered (>0.1 Hz) to eliminate DC shifts and slow drifts, and was digitally integrated with a time constant of 0.05 s to calculate breathing frequency, period, $V_T$, $T_I$, and $T_E$ during eupnea or sigh (*Appendix 1—figure 1d*). Breathing frequency is measured by the time interval between peaks. $T_I$ was calculated by the time interval from start to the maximum of the peak, $T_E$ was calculated by the time interval from the previous inspiratory peak to the onset of the next inspiratory burst and $V_T$ was calculated from the inspiratory peak. Due to the high reproducibility of effects of photostimulation, $V_T$, $T_I$, and $T_E$ during ectopic sighs were calculated as an average of 3–5 sighs, paired t-tests were used to determine statistical significance of changes before and after pharmacological injections or photostimulation. For statistical comparisons of more than two groups, repeated-measures (RM) ANOVAs were performed. For one-way RM ANOVAs, post hoc significance for pairwise comparisons was analyzed using the Holm-Sidak method. Significance was set at $p<0.05$. Data are shown as mean ± SE.

## Materials availability

Newly created materials (i.e. Grpr$^{flp}$, Nmb$^{Cre}$, and Nmbr$^{Cre}$ mice) are currently maintained at the UCLA animal facility and are available from the corresponding author, JLF, upon reasonable request.

## Acknowledgements

This work was funded by NIH Grants HL135779, NS72211 and NSFC Grant 37101012.

## Additional information

### Funding

| Funder | Grant reference number | Author |
| --- | --- | --- |
| National Heart, Lung, and Blood Institute | R35HL135779 | Jack L Feldman |
| National Heart, Lung, and Blood Institute | NS72211 | Jack L Feldman |
| National Natural Science Foundation of China | 37101012 | Yan Cui |
| National Heart, Lung, and Blood Institute | R35HL171451 | Jack L Feldman |

The funders had no role in study design, data collection and interpretation, or the decision to submit the work for publication.

### Author contributions

Yan Cui, Evgeny Bondarenko, Carolina Thörn Perez, Delia N Chiu, Conceptualization, Data curation, Formal analysis, Investigation, Visualization, Methodology, Writing – original draft, Writing – review and editing; Jack L Feldman, Conceptualization, Resources, Supervision, Funding acquisition, Writing – original draft, Project administration, Writing – review and editing

### Author ORCIDs

Yan Cui https://orcid.org/0000-0003-1336-1342
Evgeny Bondarenko https://orcid.org/0000-0003-0631-8936
Carolina Thörn Perez https://orcid.org/0000-0002-3480-8599
Delia N Chiu https://orcid.org/0000-0002-7343-7962
Jack L Feldman https://orcid.org/0000-0003-3692-9412

### Ethics

The study and experimental protocols were in accordance with the guidelines approved by the University of California, Los Angeles (UCLA) Institutional Animal Care and Use Committee and Animal Research Committee (#1994-159-83 and #2009-107).

Reviewer #1 (Public review): https://doi.org/10.7554/eLife.100192.3.sa1
Reviewer #2 (Public review): https://doi.org/10.7554/eLife.100192.3.sa2
Reviewer #3 (Public review): https://doi.org/10.7554/eLife.100192.3.sa3
Author response https://doi.org/10.7554/eLife.100192.3.sa4

## Additional files

### Supplementary files
MDAR checklist

### Data availability
Physiology data (LabChart files) are deposited to Harvard Dataverse under the accession code N0CFWQ.

The following dataset was generated:

| Author(s) | Year | Dataset title | Dataset URL | Database and Identifier |
|---|---|---|---|---|
| Bondarenko E, Cui Y, Feldman JL | 2025 | Sigh generation in preBötzinger Complex | https://doi.org/10.7910/DVN/N0CFWQ | Harvard Dataverse, 10.7910/DVN/N0CFWQ |

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

# Appendix 1

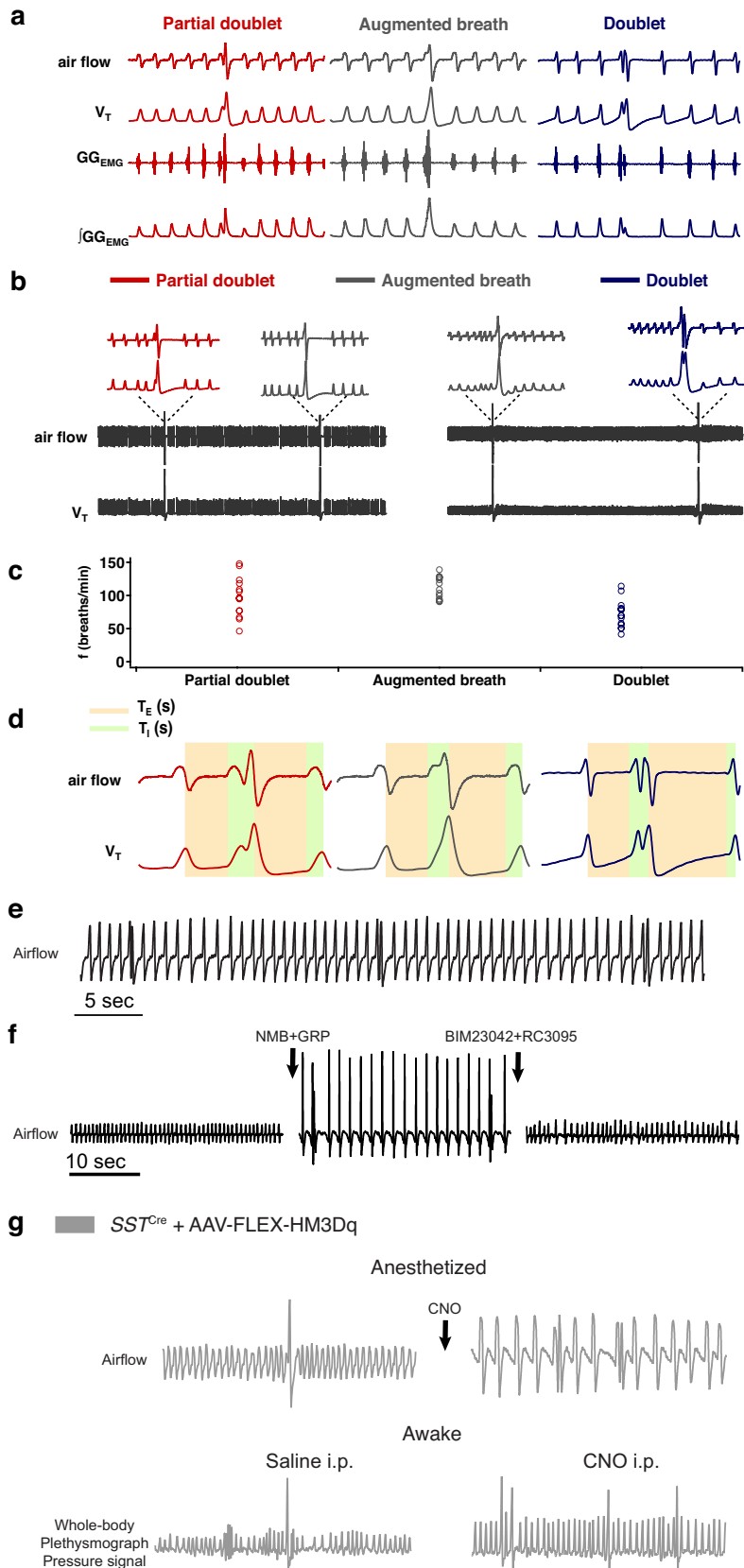

**Appendix 1—figure 1.** Characterization and types of sighs. (**a**) Three types of sighs observed in adult mice in vivo. A sigh is a spontaneous inspiratory effort that results in significantly increased inspiratory tidal volume, typically two to five times compared to normal breaths. Under anesthesia in mice, sighs in vivo can take multiple shapes. The most common shape is partial doublet, a biphasic double-sized breath with an initial phase that is identical to a normal breath (eupnea) and a later high-amplitude inspiration, coincident with a biphasic genioglossus$_{EMG}$ (GG$_{EMG}$) event (left). An in vivo sigh in mice can also present as one large breath, a monophasic augmented inspiration that has two to five times the volume of a normal breath, coincident with a monophasic GG$_{EMG}$ event (middle); or a double-peaked breath (doublets) with the first breath immediately followed by a similar amplitude second breath (right). (**b**) Raw and expanded traces show three types of shape observed in one mouse. (**c**) Sigh shapes are related to basal breathing frequency (45 spontaneous sighs from 3 mice). Different sigh shapes represent a continuum from 'augmented breath' on one side of the spectrum to 'doublet' on the other. Exact shape appears to be related to basal breathing frequency, with augmented breaths and occasional partial doublets exhibited during conditions with high basal respiratory rate/low amplitude; while doublets and partial doublets occur mostly with low basal respiratory rate/high amplitude. The latter is similar to conditions of vagotomy and in vitro, where breathing $f$ is substantially reduced, and during which sighs often appeared as doublets. (**d**) Measurement of respiratory parameters of eupneic breath and three types of sighs. (**e**) Sighs exhibiting a doublet shape occur periodically at a frequency significantly lower than normal eupneic breaths after NMB and GRP microinjection into preBötC in ketamine/xylazine anesthetized mice. (**f**) Doublets are not artifacts resulting from any damage to preBötC, since doublet-shaped sighs induced by microinjection of NMB and GRP were blocked by subsequent preBötC microinjection of NMBR and GRPR antagonists BIM23042 and RC3095. (**g**) Chemogenetic excitation of SST preBötC neurons evokes frequent doublet-shaped sighs in ketamine/xylazine mice, while in awake mice it induces frequent sighs of augmented breath shape, suggesting that doublets are indeed sighs.

