## [Editor Report · eLife Assessment]

This **valuable** study by Cui et al. investigates mechanisms generating sighs, which are crucial for respiratory function and linked to emotional states. Utilizing advanced methods in mice, they provide **solid** evidence that increased excitability in specific preBötzinger complex neuronal subpopulations expressing Neuromedin B receptors, gastrin-releasing peptide receptors, or somatostatin can induce sigh-like large amplitude inspirations. With additional technical clarifications and further elaboration of the limitations in terms of how the results are interpreted in the revised manuscript, the study will interest neuroscientists studying respiratory neurobiology and rhythmic motor systems.

---

## [Referee Report · Reviewer #1 (Public review)]

Summary of what is achieved: This manuscript validates and extends upon the sigh generating circuit between the NMB/GRP+ RTN/parafacial neurons and the NMBR/GRPR+ preBötC neurons established in Li et al., 2016. The authors generate multiple transgenic lines that enable selective targeting of these various sub-populations of cells and demonstrate the sufficiency of each type in generating a sigh breath. Additionally, they show that NMBR and GPRP preBötC neurons are glutamatergic, have overlapping and distinct expression, and do not express SST. Beyond this validation, the authors show that ectopic stimulation of SST neurons is sufficient to evoke sighs and that they are necessary for NMB/GRP induced sighing. This data is the first time that preBötC neurons downstream of NMBR/GRPR neurons have been identified that transform a eupneic breath into a sign breath. The five conclusions stated at the end of the introduction are supported by the data.

Summary of a primary weakness: A strong emphasis throughout the manuscript is the identification of an unsubstantiated slow sigh rhythm that is produced by NMBR/GRPR neurons. It is even suggested that this is an intrinsic property of these neurons. However, to make such a novel (and quite surprising) claim requires many more studies and the conclusion is dependent on how the authors have defined a sigh. Moreover, some data within the paper conflicts with this idea. The resubmitted manuscript does not contain any revisions and the rebuttal does not sufficiently address the critiques.

In summary, the optogenetic and chemogenetic characterization of the neuropeptide pathway transgenic lines nicely aligns with and provides important validation of the previous study by Li et. al., 2016 and the SST neuron studies provide a new mechanism for the transformation of NMBR/GRPR neuropeptide activation into a sigh. These are important findings, and they should be the points emphasized. The proposal of a slow sigh rhythm should be more rigorously established with new experiments and analysis or should be more carefully described and discussed.

---

## [Referee Report · Reviewer #2 (Public review)]

Summary:

This study investigates in mice neural mechanisms generating sighs, which are periodic large-amplitude breaths occurring during normal breathing that subserve physiological pulmonary functions and are associated with emotional states such as relief, stress, and anxiety. Sighs are generated by a structure called the preBötzinger complex (preBötC) in the medulla oblongata that generates various forms of inspiratory activity including sighs. The authors have previously described a circuit involving neurons producing bombesin-related peptides Neuromedin B (NMB) and gastrin releasing peptide (GRP) that project to preBötC neurons expressing receptors for NMB (NMBRs) and GRP (GRPRs) and that activation of these preBötC neurons via these peptide receptors generates sighs. In this study the authors further investigated mechanisms of sigh generation by applying optogenetic and chemogenetic strategies to selectively activate the subpopulations of preBötC neurons expressing NMBRs and/or GRPRs, and a separate subpopulation of neurons expressing somatostatin (SST) but not NMBRs and GRPRs. The authors present convincing evidence that sigh-like inspirations can be evoked by photostimulation of the preBötC neurons expressing NMBRs or GRPRs. Photostimulation of SST neurons can independently evoke sighs, and chemogenetic inhibition of these neurons can abolish sighs. The results presented support the authors' conclusion that the preBötC neurons expressing NMBRs or GRPRs produce sighs via pathways to downstream SST neurons. Thus, these studies have identified some of the preBötC cellular elements likely involved in generating sighs.

Strengths:

(1) This study employs an effective combination of electrophysiological, transgenic, optogenetic, chemogenetic, pharmacological, and neuron activity imaging techniques to investigate sigh generation by distinct subpopulations of preBötC neurons in mice.

(2) The authors extend previous studies indicating that there is a peptidergic circuit consisting of NMB and GRP expressing neurons that project from the parafacial (pF) nucleus region to the preBötC and provides sufficient input to generate sighs, since photoactivation of either pF NMB or GRP neurons evoke ectopic sighs in this study.

(3) Solid evidence is presented that sighs can be evoked by direct photostimulation of preBötC neurons expressing NMBRs and/or GRPRs, and also a separate subpopulation of neurons expressing somatostatin (SST) but not NMBRs and GRPRs.

(4) The mRNA-expression data presented from in situ hybridization indicates that most preBötC neurons expressing NMBR, GRPR (or both) are glutamatergic and excitatory.

(5) Measurements in slices in vitro indicate that only the NMBR expressing neurons are normally rhythmically active during normal inspiratory activity and endogenous sigh activity.

(6) Evidence is presented that activation of preBötC NMBRs and/or GRPRs is not necessary for sigh production, suggesting that sighs are not the unique product of the preBötC bombesin-peptide signaling pathway.

(7) The novel conclusion is presented that the preBötC neurons expressing NMBRs and/or GRPRs produce sighs via the separate downstream population of preBötC SST neurons, which the authors demonstrate can independently generate sighs, whereas chemogenetic inhibition of preBötC SST neurons selectively abolishes sighs generated by activating NMBRs and GRPRs.

Weaknesses:

(1) While these studies have identified subpopulations of preBötC neurons capable of episodically evoking sigh-like inspiratory activity, mechanisms producing the normal slow sigh rhythm were not investigated and remain unknown.

(2) The authors have addressed some of the reviewers' main technical concerns and issues relating to interpretation of the results in their rebuttal letter, but have minimally revised the manuscript. Accordingly, there remain important technical and interpretation issues requiring resolution in the revised manuscript.

Comments on revisions:

The authors have clarified in their rebuttal letter the rationale for utilizing two different photostimulation paradigms but have not incorporated any of this explanation in Methods, which would be helpful for readers.

---

## [Referee Report · Reviewer #3 (Public review)]

Summary:

This manuscript by Cui et al., studies the mechanisms for the generation of sighing, an essential breathing pattern. This is an important and interesting topic, as sighing maintains normal pulmonary function and is associated with various emotional conditions. However, the mechanisms of its generation remain not fully understood. The authors employed different approaches, including optogenetics, chemogenetics, intersectional genetic approach, and slice electrophysiology and calcium imaging, to address the question, and found several neuronal populations are sufficient to induce sighing when activated. Furthermore, ectopic sighs can be triggered without the involvement of neuromedin B (NMB) or gastrin releasing peptide (GRP) or their receptors in the preBötzinger Complex (preBötC) region of the brainstem. Additionally, activating SST neurons in the preBötC region induces sighing, even when other receptors are blocked. Based on these results, the authors concluded that increased excitability in certain neurons (NMBR or GRPR neurons) activates pathways leading to sigh generation, with SST neurons serving as a downstream component in converting regular breaths into sighs.

Strengths:

The authors employed a combination of various sophisticated approaches, including optogenetics, chemogenetics, intersectional genetic approach, and slice electrophysiology and calcium imaging, to precisely pinpoint the mechanism responsible for sigh generation. They utilized multiple genetically modified mouse lines, enabling them to selectively manipulate and observe specific neuronal populations involved in sighing.

Using genetics and calcium imaging, the authors record the neuronal activity of NMBR and GRPR neurons, respectively, and identified their difference in activity pattern. Furthermore, by applying the intersectional approach, the authors were able to genetically target and manipulate several distinct neuronal populations, such as NMBR+, GRPR- neurons and GRPR+, NMBR- neurons, and conducted a detailed characterization of their functions in influencing sighing.

Weaknesses:

(1) The authors employed two conditions for optogenetic activation: long pulse photostimulation (LPP) and short pulse photostimulation (SPP), with durations ranging from 4-10s for LPP and 100-500 ms for SPP. These could generate huge variability in the experiments. The rationale behind the selection of these conditions in each experiment remains unclear in the manuscript. Additionally, it is not explained why these specific durations were chosen. Furthermore, the interpretation for the varied responses observed under these conditions is not provided. Clarification on the rationale and interpretation of these experimental parameters would enhance the understanding of the results. The description of the experiment conditions should be consistent throughout the manuscript.

(2) Regarding the fiber optics, my understanding is that they are placed outside of the brainstem from the ventral side. Given the locations of the pF and preBötC neurons, could the differences in responses be attributed to the varying distances of each population from the ventral surface? In fact, in Figure 8, NMBR is illustrated as being closer to the ventral surface. Does it represent the actual location of these neurons?

(3) The results of recording on NMBR neurons in Figure 4 were compelling. However, I'm curious why the recording of GRPR neurons and their response to the neuropeptide were not presented or examined. Additionally, considering the known cross-reaction between peptides and their receptors, it might be worthwhile to investigate how GRP modulates NMBR neurons and how NMB modulates GRPR neurons.

(4) The authors found that activation of several preBötC populations, including NMBR, GRPR, and SST neurons, despite pharmacological inhibition of NMBR and GRPR, can still induce sighing, and concluded that "activation of preBötC NMBRs and/or GRPRs is not necessary for sigh production". I disagree with this conclusion. Even when the receptors are silenced, artificial (optogenetic or chemogenetic) activation could still activate the same downstream pathways. This cannot be used as evidence to claim that the receptors are not required for sighing in vivo, because it is possible that the receptors are still necessary for the activation of these neurons under natural conditions. For instance, while diaphragm activation induces breathing, it does not negate the crucial role of the nervous system in regulating this process in physiological conditions.

(5) The authors noted varied responses upon activating specific subpopulations of the preBötC neurons, namely NMBR, GRPR, and SST neurons. Could these differences be attributed to variations in viral labeling efficiency among different mouse genetic lines? Are there discrepancies in the number of labeled neurons across the lines? Additionally, the authors did not thoroughly characterize the specificities of AAV targeting in their Cre and Flp lines. It's uncertain whether the AAV-labeled neurons are strictly restricted to the designated population without notable leakage into other populations. This is particularly crucial for the experiments manipulating SST neurons. If there's substantial labeling of NMBR or GRPR neurons, it could undermine the conclusions drawn. Further examination of the precision and selectivity of the labeling techniques is necessary to ensure the accurate interpretation of the experimental findings.

(6) The authors have addressed some of the reviewers' concerns in the revision; however, many important issues remain unaddressed.

---

## [Author Response]

The following is the authors’ response to the original reviews.

(1) Reviewer 3: Moreover, the conclusion that preBötC NMBR and GRPR activations are unnecessary for sighing is not fully supported by the current experimental design. While the study shows that sighing can still be induced despite pharmacological inhibition of NMBR and GRPR, this does not conclusively prove that these receptors are not required under natural conditions.

We concluded that “NMBR and GRPR receptors are not necessary for sigh generation”. We acknowledge that under normal conditions these receptors almost certainly play a role; in fact, microinjection of saporin conjugated to bombesin, which presumably ablates NMBR^+^ and GRPR^+^ preBötC neurons, completely eliminated endogenous sighing activity in awake mice (Li et al., Nature, 2015). However, that study did not establish that the receptors *per se* are essential in this context, since the protocol ablated not just the receptors but also the preBötC neurons that happened to express these receptors. Here, we show that we could evoke sighs AFTER complete pharmacological blockade of NMBRs *and* GRPRs. Also, we show that sighs can be elicited by stimulation of a distinct subpopulation of preBötC neurons expressing the peptide somatostatin (SST^+^). These results demonstrate that sighs can be evoked in absence of activation of NMBRs and/or GRPRs, leading to the conclusion that NMBRs and/or GRPRs are not required for sighs but rather contribute to periodic sigh generation under normal conditions.

(2) Reviewer 1: To make such a novel (and quite surprising) claim requires many more studies and the conclusion is dependent on how the authors have defined a sigh. Moreover, some data within the paper conflicts with this idea.

Our definition of sighs was carefully chosen so that it applied across different experimental conditions, including in vitro slices, anesthetized or awake in vivo. We defined sighs as transient changes in minute ventilation on a time scale slower than eupneic breathing period, to avoid classifying breathing after vagotomy or under isoflurane anesthesia as “all-sigh breathing”. This is why induction of persistent large amplitude breaths (such as in Figures 5-6) were not counted as sighs.

(3) Reviewer 2: Several key technical aspects of the study require further clarification to aid in interpreting the experimental results, including issues relating to the validation of the transgenic mouse lines and virally transduced expressions of proteins utilized for optogenetic and chemogenetic experiments, as well as justifying the optogenetic photostimulation paradigms used to evoke sighs.

The rationale for using SPP and LPP stems from our published observations of the effects of optogenetic stimulation of various preBötC neuronal subpopulations. Thus, SPP and LPP evoke the same responses in GlyT2 (Sherman et al., 2015) and Dbx1 (Cui et al., 2016) neurons, while for other subpopulations, e.g., SST (Cui et al., 2015), the effects of SPP are markedly different from LPP. Hence, in this study we examined both. As effects of SPP and LPP of SST neurons were examined previously (Cui et al., 2016), these protocols were not repeated except for evoking sighs after blockade of NMBR/GRPRs. SPP of pF NMB or GRP did not evoke any respiratory responses and hence were not presented in any figures (see Results, section “Activation of Nmb- or Grp-expressing pF neurons induces sighs”).

(4) Reviewer 3: however, the rationale and experimental details require further explanation, and their impacts on the conclusion require clarification. For instance, how and why the variability in optogenetic activation conditions could impact the experimental outcomes.

Refractory periods reported here for pF NMB, pF GRP, preBötC NMBR and preBötC GRPR were all obtained using the same intensity LPP. We acknowledge the possibility, even the likelihood that higher intensity LPP would shorten refractory periods. In line with this, we observed that ectopic sighs were evoked earlier during the LPP as the sigh phase progressed. As described in RESULTS, such effects were observed for pF NMB, pF GRP, preBötC NMBR and preBötC GRPR only and not for preBötC SST, which might suggest that timing of intrinsically generated sighs depends on the NMB-GRP signaling pathway, yet sigh production depends on the SST pathway.